# Glucocorticoid receptor signalling activates YAP in breast cancer

Giovanni Sorrentino[1,*], Naomi Ruggeri[1,*], Alessandro Zannini[1], Eleonora Ingallina[1,2], Rebecca Bertolio[1], Carolina Marotta[1], Carmelo Neri[1,2], Elisa Cappuzzello[3], Mattia Forcato[4], Antonio Rosato[3,5], Miguel Mano[6,7], Silvio Bicciato[4] & Giannino Del Sal[1,2]

The Hippo pathway is an oncosuppressor signalling cascade that plays a major role in the control of cell growth, tissue homoeostasis and organ size. Dysregulation of the Hippo pathway leads to aberrant activation of the transcription co-activator YAP (Yes-associated protein) that contributes to tumorigenesis in several tissues. Here we identify glucocorticoids (GCs) as hormonal activators of YAP. Stimulation of glucocorticoid receptor (GR) leads to increase of YAP protein levels, nuclear accumulation and transcriptional activity *in vitro* and *in vivo*. Mechanistically, we find that GCs increase expression and deposition of fibronectin leading to the focal adhesion-Src pathway stimulation, cytoskeleton-dependent YAP activation and expansion of chemoresistant cancer stem cells. GR activation correlates with YAP activity in human breast cancer and predicts bad prognosis in the basal-like subtype. Our results unveil a novel mechanism of YAP activation in cancer and open the possibility to target GR to prevent cancer stem cells self-renewal and chemoresistance.

[1] Laboratorio Nazionale CIB (LNCIB), Area Science Park Padriciano, Trieste 34149, Italy. [2] Dipartimento di Scienze della Vita, Università degli Studi di Trieste, Trieste 34127, Italy. [3] Department of Surgery, Oncology and Gastroenterology, University of Padova, Padova 35124, Italy. [4] Department of Life Sciences, University of Modena and Reggio Emilia, Modena 41125, Italy. [5] Veneto Institute of Oncology IOV-IRCCS, Padova 35128, Italy. [6] Center for Neuroscience and Cell Biology (CNC), University of Coimbra, Coimbra 3060-197, Portugal. [7] International Centre for Genetic Engineering and Biotechnology (ICGEB), Trieste 34149, Italy. * These authors contributed equally to this work. Correspondence and requests for materials should be addressed to G.D.S. (email: delsal@lncib.it).

Cell proliferation, tissue growth and organ size control are primarily regulated by the Hippo signalling, a pathway originally discovered in *Drosophila melanogaster*, and highly conserved in higher eukaryotes[1]. The core of the mammalian Hippo pathway is a serine/threonine kinase cassette represented by the Ste20-like MST1/2 (Hippo in *Drosophila*) kinases and the large tumour suppressor LATS1/2. Activated MST1/2 in association with the adaptor protein Salvador (Sav1) phosphorylate and activate the downstream kinases LATS1/2, which in turn phosphorylate and inhibit the Hippo pathway nuclear effectors YAP (Yes-associated protein) and TAZ (transcriptional co-activator with PDZ-binding motif). While phosphorylated YAP/TAZ are retained in the cytoplasm and directed toward proteasome-dependent degradation, dephosphorylated YAP/TAZ localize in the nucleus and function as transcription co-activators for the TEAD family of transcription factors to induce gene expression thereby promoting the cell growth, proliferation and survival[2,3].

Disruption of Hippo signalling in mouse models promotes tumour formation, while transgenic YAP overexpression in mice results in hyperplasia and eventual tumour development[4,5]. Aberrant activation of YAP and TAZ has been observed in many human cancers, including high-grade and metastatic breast cancer, and it is associated with tumour initiation, progression and metastasis[6,7]. The YAP/TAZ-induced transcriptional program leads to an increase in cell proliferation, migration, self-renewal of cancer stem cells (CSCs), epithelial-to-mesenchymal transition and drug resistance, making these transcription co-activators attractive targets for cancer therapies[1,3].

While the core components of the Hippo pathway are well established, a number of additional upstream regulators are emerging. YAP/TAZ, in fact, have turned out to be primary sensors of a variety of signals generated on cell–cell contacts, by adhesion and apical–basal polarity proteins, by the mechanical cues of the neighbouring cells and the extracellular matrix (ECM), as well as in relation to the metabolic state of the cell[2,3,8]. Moreover, a number of diffusible extracellular ligands have been recently entered into the list of YAP/TAZ regulators. In this context, it has been demonstrated that some signalling molecules, among which lysophosphatidic acid and epinephrine, control YAP/TAZ via G-protein-coupled receptors[9]. Also oestrogens have been shown to strongly impact on YAP/TAZ regulation, but only through the membrane-associated G-protein-coupled oestrogen receptor[10]. However open questions remain on whether YAP/TAZ are controlled by hormones acting through nuclear receptors.

Here, we identify glucocorticoids (GCs) as hormonal activators of YAP in breast cancer cells. We demonstrate that GCs treatment promotes fibronectin deposition, focal adhesion-dependent activation of Src and the remodelling of the actin cytoskeleton, which ultimately results in sustained YAP activity. Moreover, we find that in breast cancer cells the activation of glucocorticoid receptor (GR) is fundamental for CSCs self-renewal and chemoresistance and that this effect is the consequence of the transcriptional activity of activated YAP. Our results unveil an unpredicted layer of YAP regulation and put the GR–YAP axis at the cornerstone of a potential new therapeutic strategy to specifically target CSCs in breast cancer.

## Results

### GCs induce activation of the Hippo transducer YAP.
To identify novel signalling pathways able to control YAP activation in breast cancer we performed a high-content, fluorescence microscopy-based, high-throughput screening using a library of FDA: Food and Drug Administration (FDA)-approved drugs composed of a collection of 640 clinically used compounds with known and well-characterized bioactivity, safety and bioavailability[11]. The activation of YAP observed in human cancers is linked to increased levels of YAP protein that accumulate in the nucleus[12]. Therefore, we monitored the effect of each compound of the library on YAP protein levels in the triple-negative breast cancer cell line MDA-MB-231. Drugs were added to the culture medium for 24 h and total YAP protein amount was detected by immunofluorescence followed by automated quantification of the YAP-relative signal at single-cell level (Supplementary Fig. 1a). With this analysis, we identified drugs that either reduced or increased YAP protein amount (Fig. 1a). Interestingly, the compounds with the strongest effect in increasing YAP protein levels belong to the class of synthetic GCs (Fig. 1a,b). Betamethasone (BM) was among the strongest hits and, together with hydrocortisone (HC) and dexamethasone (DM), was selected for further analysis.

Natural GCs, such as cortisol, are a class of steroid hormones acting through specific nuclear receptors and playing important roles in various physiological processes, such as metabolism, immune response and development[13]. Owing to their anti-inflammatory and immunosuppressive actions, synthetic GCs have been widely used in the treatment of inflammatory and autoimmune diseases[14].

To evaluate the response of MDA-MB-231 cells on GCs stimulation, we used the GCs-responsive reporter MMTV-luc[15]. Treatment with BM for 24 h led to strong increase of luciferase signal, which was completely prevented by concomitant administration of the GC and progesterone receptors inhibitor RU486 (Supplementary Fig. 1b). To validate the results of the screening we monitored YAP protein levels by western blot (WB) in MDA-MB-231 cells, in Ras-transformed MCF10A-T1k (hereafter MII) cells and in immortalized normal mammary MCF10A cells on treatment with GCs[16]. BM led to a GR-dependent increase of YAP protein levels in all the investigated cell lines (Fig. 1c), an effect not observed for the YAP paralog TAZ (Supplementary Fig. 1c).

In cells, YAP acts mainly as TEADs transcription co-factor inducing the expression of several genes[17]. To determine whether the GCs-induced increase of YAP levels was associated with its functional activation, we monitored the expression of YAP target genes (*Ankrd1*, *Cyr61* and *Ctgf*) in MDA-MB-231 cells, on administration of GCs[18]. Interestingly, along with increased expression of GILZ, a well known GC-responsive gene[19], BM treatment increased the expression of YAP target genes and this effect was prevented by RU486 co-treatment (Fig. 1d,e). Moreover, short interfering RNA (siRNA)-mediated knockdown of YAP, but not of TAZ, prevented the induction of YAP targets, while having no effects on GILZ expression (Fig. 1d,e; Supplementary Fig. 1d,e). To investigate whether GCs could regulate YAP transcriptional activity *in vivo*, we systemically injected DM in C57Bl/6 mice and after 16 h we measured YAP transcriptional activity in the mammary tissue. We used the expression levels of YAP target genes *Ccnd1* and *Ctgf* as reporters of YAP activity, as they have been previously used with success to monitor YAP activation *in vivo* and have been reported to be expressed in breast tissue[20,21]; GILZ was used as a control for GR activation. Interestingly, BM treatment led to significant increase in *Ccnd1* and *Ctgf* expression in the mammary tissue (Fig. 1f). Taken together, these results support the notion that GCs activate YAP in mammary epithelial cells *in vitro* and *in vivo*.

### GR controls YAP in breast cancer cells.
Natural GCs are cholesterol-derived hormones released in the serum. Their

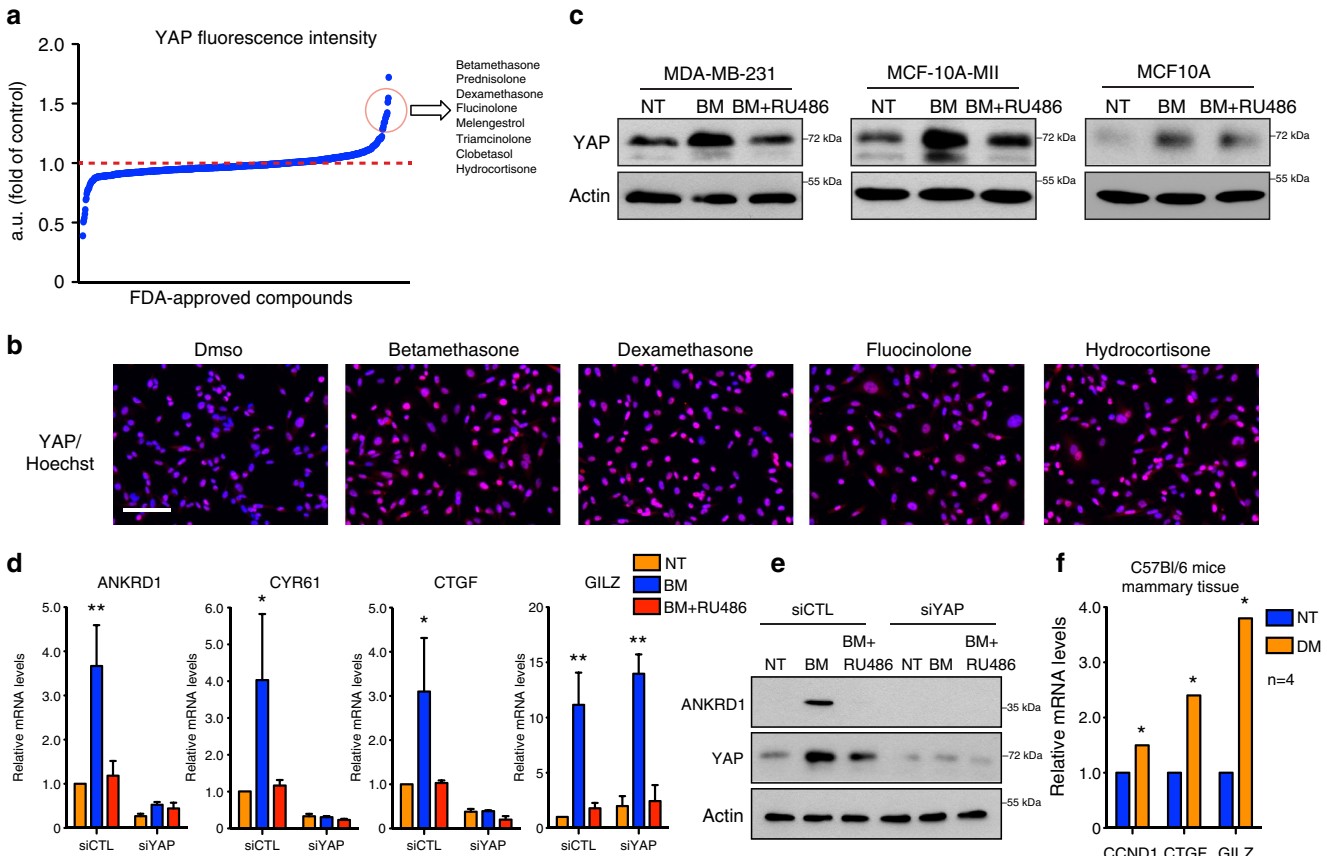

**Figure 1 | Glucocorticoids induce activation of the Hippo transducer YAP *in vitro* and *in vivo*.** (**a**) Results of the high-content screening. YAP fluorescence intensity is relative to DMSO-treated samples. (**b**) Representative images from the screening. MDA-MB-231 stained for Hoechst (blue) and YAP (red) after treatment with DMSO or the indicated glucocorticoids are shown. Experiment repeated two times. Scale bars, 100 μm. (**c**) MDA-MB-231, MCF10A and MII cells were treated with BM 1 μM alone or in combination with RU486 1 μM for 24 h. Representative blots are shown. Experiment repeated three times. (**d**) Quantitative PCR with reverse transcription (qRT–PCR) analysis of MDA-MB-231 transfected with indicated siRNA for 48 h and treated with 1 μM BM alone or in combination with RU486 1 μM for 24 h. siCTL is control siRNA. Error bars represent mean ± s.d., from n = 3 biological replicates. siYAP sequence is siYAP#1. (**e**) MDA-MB-231 cells were treated as in **d**, representative blots are shown. Experiment repeated three times. siYAP sequence is siYAP#1. (**f**) qRT–PCR analysis of breast epithelial tissue from control (NT) or dexamethasone (DM)-treated mice. Error bars represent mean ± s.d., n = 4 mice per group. *P < 0.05, **P < 0.01; two-tailed Student's t-test is used throughout.

synthesis and relapse are under circadian and stress-associated regulation exerted by the hypothalamic–pituitary–adrenal axis[22]. Both natural and synthetic GCs mediate their effects on target cells by binding to GR and inducing its translocation into the nucleus, where it acts as a transcription factor that positively or negatively regulates the expression of GC-responsive genes[22]. Several soluble serum factors, such as G-protein-coupled receptor agonists, cytokines and growth factors, have been shown to control YAP activity in cancer cells[2]; thus, we hypothesized that, acting through GR, GCs may represent a novel class of serum regulators of YAP. To test this hypothesis, we first assessed the ability of serum GCs to activate endogenous GR in breast cancer cells. As expected, in serum-starved MDA-MB-231 cells GR localized mainly in the cytoplasm in an inactive state (Fig. 2a). On addition of 10% fetal bovine serum (FBS) to cells for 24 h a clear nuclear localization of GR was observed in almost 50% of the cells while, after BM treatment, all the cells (100%) scored positive (Fig. 2a; Supplementary Fig. 1f). To assess whether serum GCs control YAP activation in breast cancer cells, we knocked down GR by siRNA transfection in MDA-MB-231, BT-549 and MII cells grown in 10% FBS. Interestingly, GR silencing led to a strong reduction of the protein levels of YAP, but not of TAZ, and this effect was due to increased the

protein degradation (Fig. 2b; Supplementary Fig. 1g,h). Moreover, GR silencing reduced YAP transcriptional activation, as assayed by monitoring the reporter activity of a YAP-responsive 8XGTII-luc construct and the expression levels of YAP target genes (Fig. 2c,d)[11,18]. Altogether these results demonstrate that GR regulates YAP protein levels and transcriptional activity in breast cancer cells.

**GCs induce YAP nuclear localization.** The analysis of YAP subcellular localization showed that all the GCs of the drug collection used for the screening increased the percentage of cells displaying nuclear-localized YAP (Fig. 3a)[11]. Moreover, validation experiments confirmed that BM induces YAP nuclear localization in serum-starved cells (Fig. 3b,c; Supplementary Fig. 2a)[9]. As YAP activity is tightly controlled through its phosphorylation-dependent nucleocytoplasmic shuttling[6,23], we next monitored the levels of YAP phosphorylation on Ser-127, a post-translational modification associated to YAP cytoplasmic retention[23]. Interestingly, on BM treatment, MDA-MB-231 and MII cells displayed a strong decrease of YAP phosphorylation on Ser-127 along with increased YAP protein levels (Fig. 3d)[9]. In line with this, BM treatment rescued YAP nuclear localization

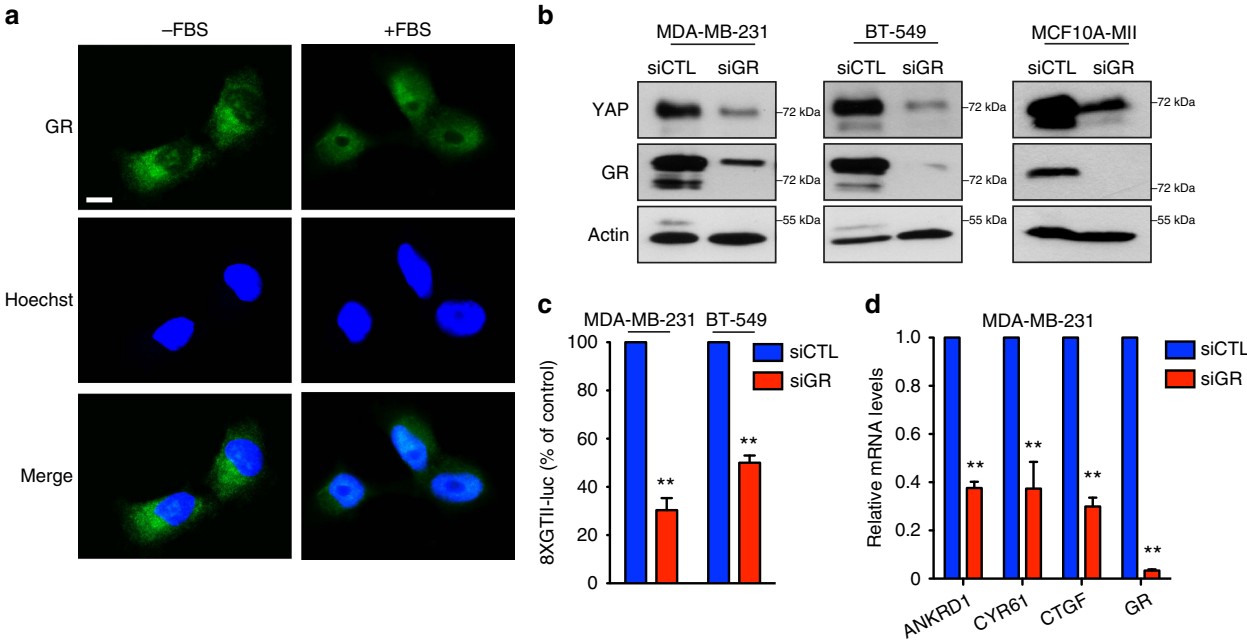

**Figure 2 | Glucocorticoid receptor controls YAP activity in breast cancer cells.** (**a**) Representative images of immunofluorescence in MDA-MB-231 cells. Cells were serum starved or grown in 10% FBS for 24 h. Experiment repeated three times. Scale bars, 15 μm. (**b**) MDA-MB-231, BT-549 and MII cells were transfected with control (siCTL) or glucocorticoid receptor (siGR) siRNA for 48 h. Representative blots are shown. Experiment repeated three times. (**c**) MDA-MB-231 and BT-549 cells were transfected with control (siCTL) or glucocorticoid receptor (siGR) siRNA. The day after, cells were transfected with 8XGTII-luc reporter. After 24 h cells were collected. Data are normalized to siCTL. Error bars represent mean ± s.d., from $n = 3$ biological replicates. (**d**) Quantitative PCR with reverse transcription analysis of MDA-MB-231 transfected with control (siCTL) or glucocorticoid receptor (siGR) siRNA for 48 h. Error bars represent mean ± s.d., from $n = 3$ biological replicates. *$P < 0.05$, **$P < 0.01$; two-tailed Student's t-test is used throughout.

in cells grown at high confluence, a condition known to inhibit YAP nuclear localization in a phosphorylation-dependent manner (Fig. 3e; Supplementary Fig. 2b).

The LATS1/2 kinases are key components of the Hippo pathway and are known to phosphorylate YAP on several residues, including Ser-127. Thus, we sought to determine whether LATS1/2 were involved in the regulation of YAP localization by GCs. Of note, while RU486 treatment induced YAP cytoplasmic localization in BM-treated cells, LATS1/2 knockdown completely prevented this effect (Fig. 3f; Supplementary Fig. 2c). To confirm the role of LATS1/2 in BM-induced YAP nuclear localization, we transiently transfected MDA-MB-231 cells with a construct expressing a mutant form of YAP (YAP-5SA) able to escape LATS1/2-mediated phosphorylation[24]. YAP-5SA was almost completely insensitive to RU486 treatment in BM-treated cells, thus suggesting that GCs regulate YAP subcellular localization in a LATS1/2-dependent manner (Supplementary Fig. 2d). Consistently, in parallel with a decrease in YAP phosphorylation, BM caused a reduction of the phosphorylated and active form of LATS1 (Fig. 3g), confirming that GCs inhibit the Hippo pathway activation.

**GR activates YAP by inducing FN1.** To explore the molecular mechanisms underlying YAP activation in response to GR stimulation, we monitored the time course of GR and YAP activation after BM treatment using GILZ (target of GR) and Ankrd1 (target of YAP) messenger RNA (mRNA) levels as read outs. As shown in Fig. 4a,b, GR nuclear translocation significantly anticipated YAP nuclear accumulation. Moreover, analysis of GILZ and Ankrd1 mRNA levels showed that GR and YAP are not concurrently activated by BM treatment (Fig. 4c). These results, together with the evidence that YAP mRNA levels are not

significantly affected by GCs (Supplementary Fig. 3c), suggest that YAP is not a direct target of GR and are consistent with a scenario, in which YAP is activated as an indirect consequence of the GR transcriptional activity. To identify GR target genes involved in the GC-dependent activation of YAP, we examined the genes induced in MDA-MB-231 cells on DM treatment as annotated in published data set[25] and we found an enrichment in genes involved in cellular adhesion (Supplementary Tables 1, 4 and 5)[26]. We focused our attention on fibronectin 1 (FN1), a glycoprotein that is essential for establishing cell adhesion to ECM and spreading (Supplementary Table 5). Recently, cell adhesion to fibronectin has been found to regulate Hippo signalling and YAP activation through mechanical stimulation of cells[27]. We also analysed the GR chromatin immunoprecipitation (ChIP)-Seq data set of Dex-stimulated A549 cells derived from the ENCODE project and we identified and validated a stringent ChIP-Seq peak in the FN1 promoter, supporting that FN1 is a direct target of GR that could activate YAP by reinforcing cell adhesion to ECM (Supplementary Fig. 3a,b). To test this hypothesis, we monitored FN1 mRNA levels in MDA-MB-231 cells after BM treatment. Of note, BM induced a strong increase of FN1 mRNA and protein levels, as well as its extracellular deposition, in a GR-dependent manner (Fig. 4d–f; Supplementary Fig. 3d,e). In line with increased fibronectin expression, microscopy analysis confirmed that cells receiving BM underwent marked morphological changes with a clear increase in cell size and spreading (Fig. 4g). These results strongly suggest that GR-induced FN1 expression and deposition might alter the ECM composition in the local microenvironment with a consequent impact on cell adhesion which, in turn, could lead to the YAP activation[27]. To test this hypothesis, we treated cells with BM in combination with a synthetic peptide (the Arg -Gly -Asp (RGD) containing peptide) that competing with ECM

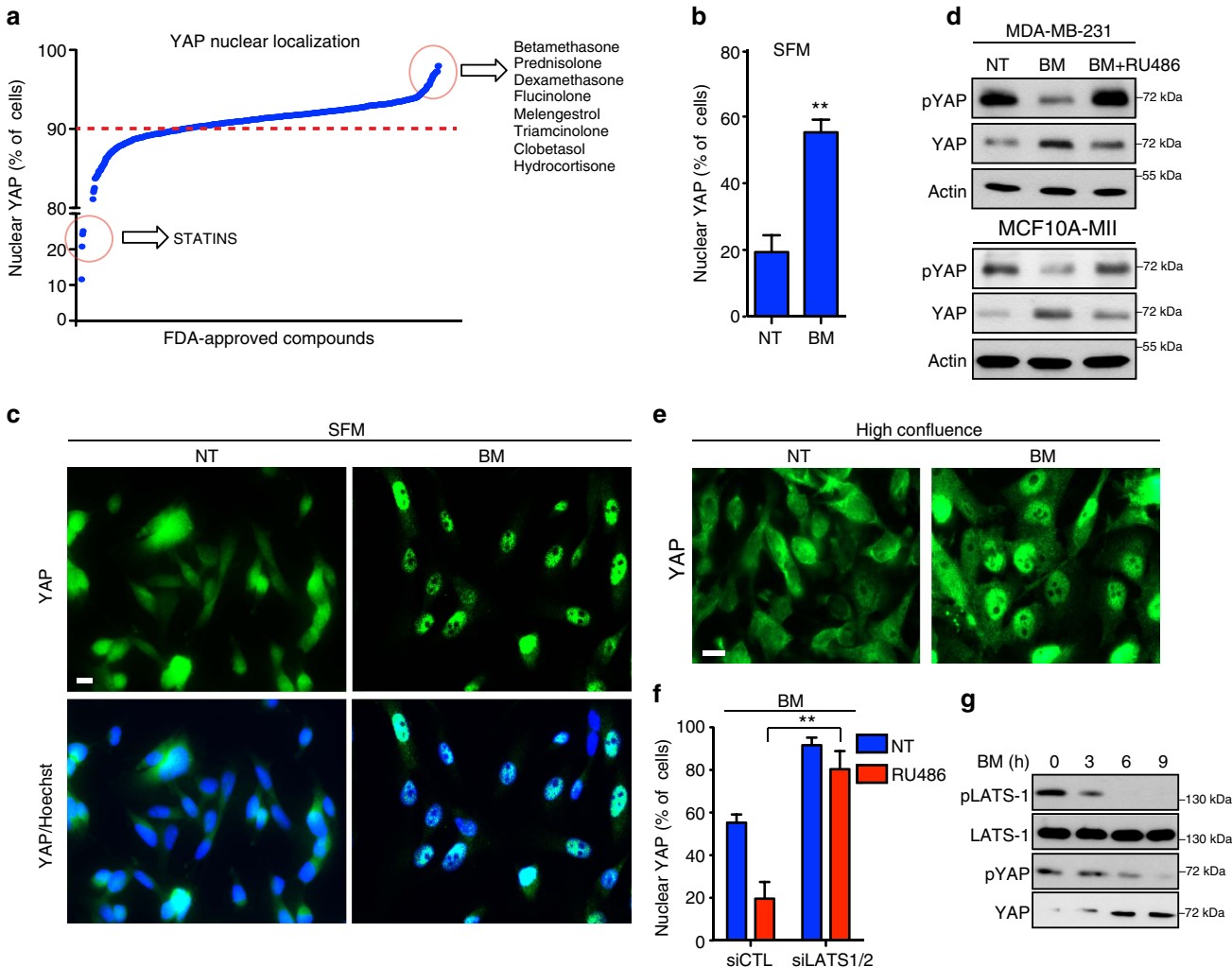

**Figure 3 | Glucocorticoids induce YAP nuclear localization.** (**a**) Results of the high-content screening. (**b**) Quantification of MDA-MB-231 cells with nuclear YAP by immunofluorescence. Cells were grown in serum-free medium (SFM) and treated with betamethasone (BM) 0.1 μM for 24 h. Error bars represent mean ± s.d., from $n = 3$ biological replicates. (**c**) Cells were treated as in **b**. Representative images are shown. Scale bars, 15 μm. (**d**) MDA-MB-231 and MII cells were grown in SFM and treated with BM 1 μM alone or in combination with RU486 1 μM for 24 h. Representative blots are shown. pYAP is phospho-Ser127. Experiment repeated three times. (**e**) MDA-MB-231 cells were grown at high confluence and treated with BM 1 μM for 24 h. Representative images are shown. Experiment repeated three times. Scale bars, 15 μm. (**f**) Quantification of cells with nuclear YAP by immunofluorescence. MDA-MB-231 cells were transfected with indicated siRNAs for 24 h and treated with RU486 1 μM for additional 24 h in SFM containing BM 1 μM. Error bars represent mean ± s.d., from $n = 3$ biological replicates. (**g**) MDA-MB-231 cells were serum starved and treated with BM 1 μM for indicated times. Representative blots are shown. *$P < 0.05$, **$P < 0.01$; two-tailed Student's $t$-test is used throughout.

proteins' binding to integrin—impedes the adhesion of cells to ECM. Of note, RGD co-treatment prevented YAP dephosphorylation, nuclear translocation and Ankrd1 upregulation induced by BM (Fig. 4h–j). In line with this evidence, FN1 and integrin αV siRNA transfection prevented the YAP activation downstream of GR stimulation (Supplementary Fig. 3f,g), demonstrating that FN1 acts as an effector of GR signalling to stimulate YAP function.

**GCs activate YAP via actin cytoskeleton remodelling.** Increased matrix stiffness is a feature of several solid tumours, and the integrated cooperation of mechanical and diffusible signals is required to maintain this physical property within the tumour mass[18,28,29]. YAP has been described as a sensor and transducer of mechanical stimuli. Indeed, in cells grown on high-rigidity substrate under high mechanical tensions generated by

actin stress fibres YAP localizes in the nucleus and activates its transcriptional program[18,27]. In this context, tension imposed by integrin-mediated cell adhesion to fibronectin leads to FAK/Src-mediated cytoskeleton rearrangement, cell spreading and LATS1/2-dependent YAP nuclear localization[27,30,31]. Moreover, actin cytoskeleton and Src functions are required for the YAP activation by stiff matrices[32]. Thus, it is conceivable that GC-induced YAP activation could be triggered by the increased deposition of fibronectin, which in turn stimulates the intracellular integrin–FAK–Src signalling. To test this hypothesis, we analysed these cascade of events by staining focal adhesions, by vinculin immunofluorescence in BM-treated MDA-MB-231 cells. As shown in Fig. 5a, along with the induction of YAP nuclear localization, BM treatment elicited a marked increase of the number of focal adhesions, which was completely prevented by the GR inhibitor RU486 (Supplementary Fig. 4a). Consistently, BM-induced Src

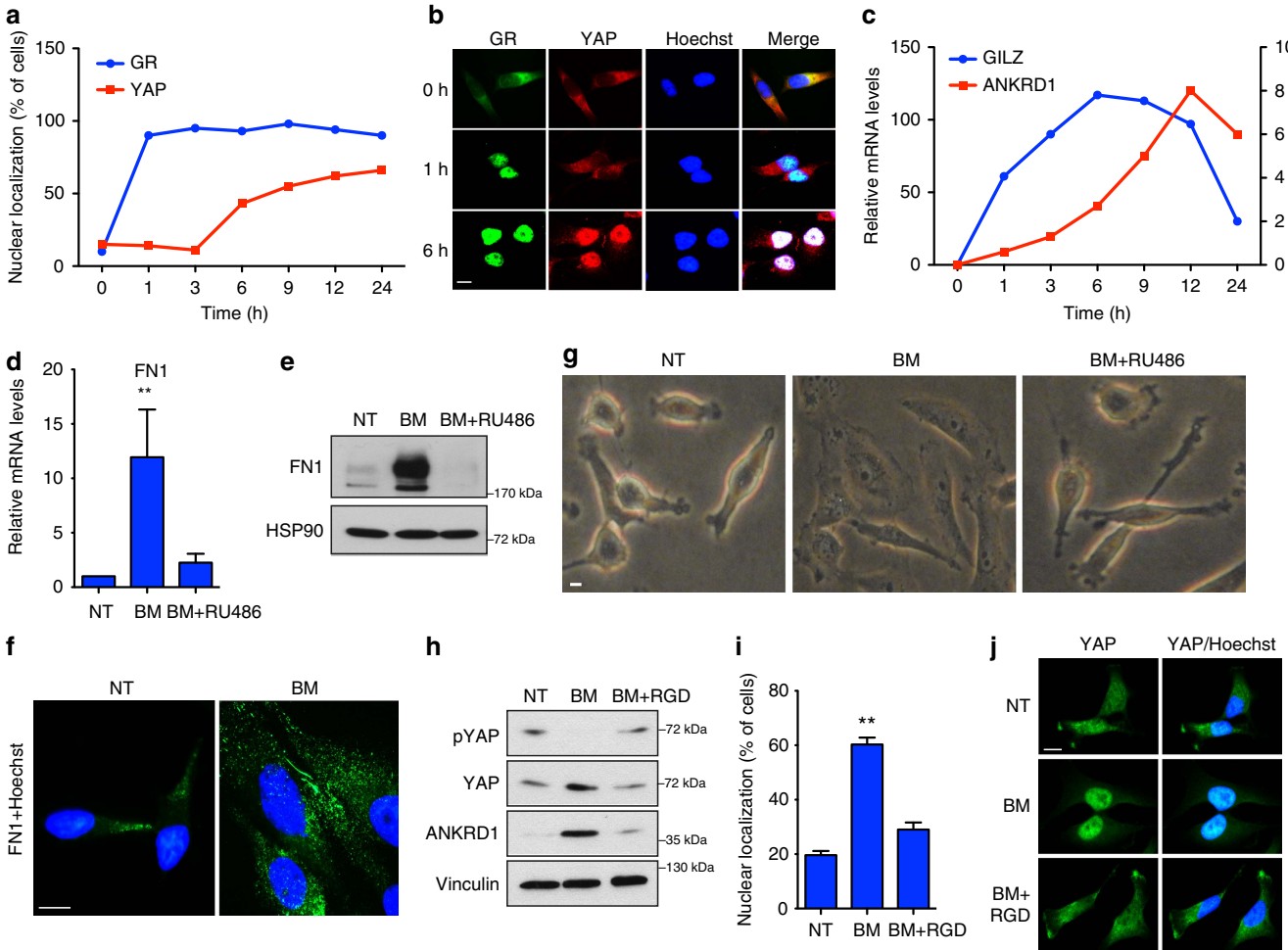

**Figure 4 | Glucocorticoid receptor activates YAP by inducing FN1 expression.** (**a,b**) Quantification of cells with nuclear YAP (**a**) and representative immunofluorescence (**b**) of serum-starved MDA-MB-231 treated with 1 μM betamethasone (BM) for the indicated times. (**c**) Quantitative PCR with reverse transcription (qRT–PCR) analysis of serum-starved MDA-MB-231 treated with 1 μM BM for the indicated times. Left y axis is relative to GILZ, right y axis is relative to ANKRD1. (**d**) qRT–PCR analysis of serum-starved MDA-MB-231 treated with 1 μM BM alone or in combination with RU486 1 μM for 24 h. Error bars represent mean ± s.d., from $n = 3$ biological replicates. (**e**) MDA-MB-231 cells were treated with BM 1 μM alone or in combination with RU486 1 μM for 24 h. Representative blots are shown. Experiment repeated three times. (**f**) MDA-MB-231 cells were serum starved and treated with BM 1 μM for 24 h. Extracellular fibronectin was stained by immunofluorescence. Experiment repeated three times. Scale bars, 15 μm. (**g**) Bright-field microscopy image of MDA-MB-231 cells treated with BM 1 μM alone or in combination with RU486 1 μM for 24 h. Experiment repeated three times. Scale bars, 15 μm. (**h**) MDA-MB-231 were serum starved and treated with BM 1 μM alone or in combination with RU486 1 μM for 6 h. Representative blots are shown. pYAP is phospho-Ser127. Experiment repeated three times. (**i,j**) Quantification (**i**) and representative images (**j**) of MDA-MB-231 cells with nuclear YAP by immunofluorescence. Cells were grown in serum-free medium and treated with BM 1 μM alone or in combination with RGD 500 μg ml$^{-1}$ for 6 h. Scale bars, 15 μm. Error bars represent mean ± s.d., from $n = 3$ biological replicates. $*P < 0.05$, $**P < 0.01$; two-tailed Student's t-test is used throughout.

activation, as demonstrated by increased phosphorylation of Src (on Y-416) and of FAK (on Y-925, a Src-specific phosphorylation site; Fig. 5b; Supplementary Fig. 4b). To prove the involvement of the focal adhesions and of Src pathway activation in YAP nuclear localization downstream of GR, we pharmacologically inhibited Src activation in BM-treated cells. Strikingly, the nuclear translocation and activation of YAP induced by BM was completely prevented by the two Src inhibitors dasatinib and saracatinib, in a LATS1/2-dependent manner (Fig. 5c–f). Similarly, treatment with a FAK inhibitor (PF573228) or growing cells in suspension prevented YAP activation on GC treatment (Fig. 5c–e; Supplementary Fig. 4c). Interestingly, while BM-induced LATS1 inhibition, which was prevented by both Src and FAK inhibition, MST1/2 activity remained unchanged after these treatments (Supplementary Fig. 4f). These results

demonstrate that, downstream of GR stimulation, Src signalling is required for YAP nuclear translocation and activation.

BM treatment also caused a strong increase in the number of F-actin stress fibres, which are downstream of Src (Fig. 5g; Supplementary Figs 4d and 5). Notably, inhibition of actin polymerization by latrunculin A treatment prevented YAP activation induced by BM, leaving totally unaffected GR activation (Fig. 5h–j; Supplementary Figs 4e and 5)[33]. Moreover, RGD, dasatinib and PF573228 treatments, as well as ablation of FN1 or intergrin αV by siRNA transfection prevented stress fibres formation downstream of GR stimulation (Supplementary Figs 3f,g and 5). Taken together, these data clearly demonstrate that on GR stimulation, activation of focal adhesion signalling caused by increased FN1 deposition leads to YAP activation via Src-dependent actin cytoskeleton remodelling.

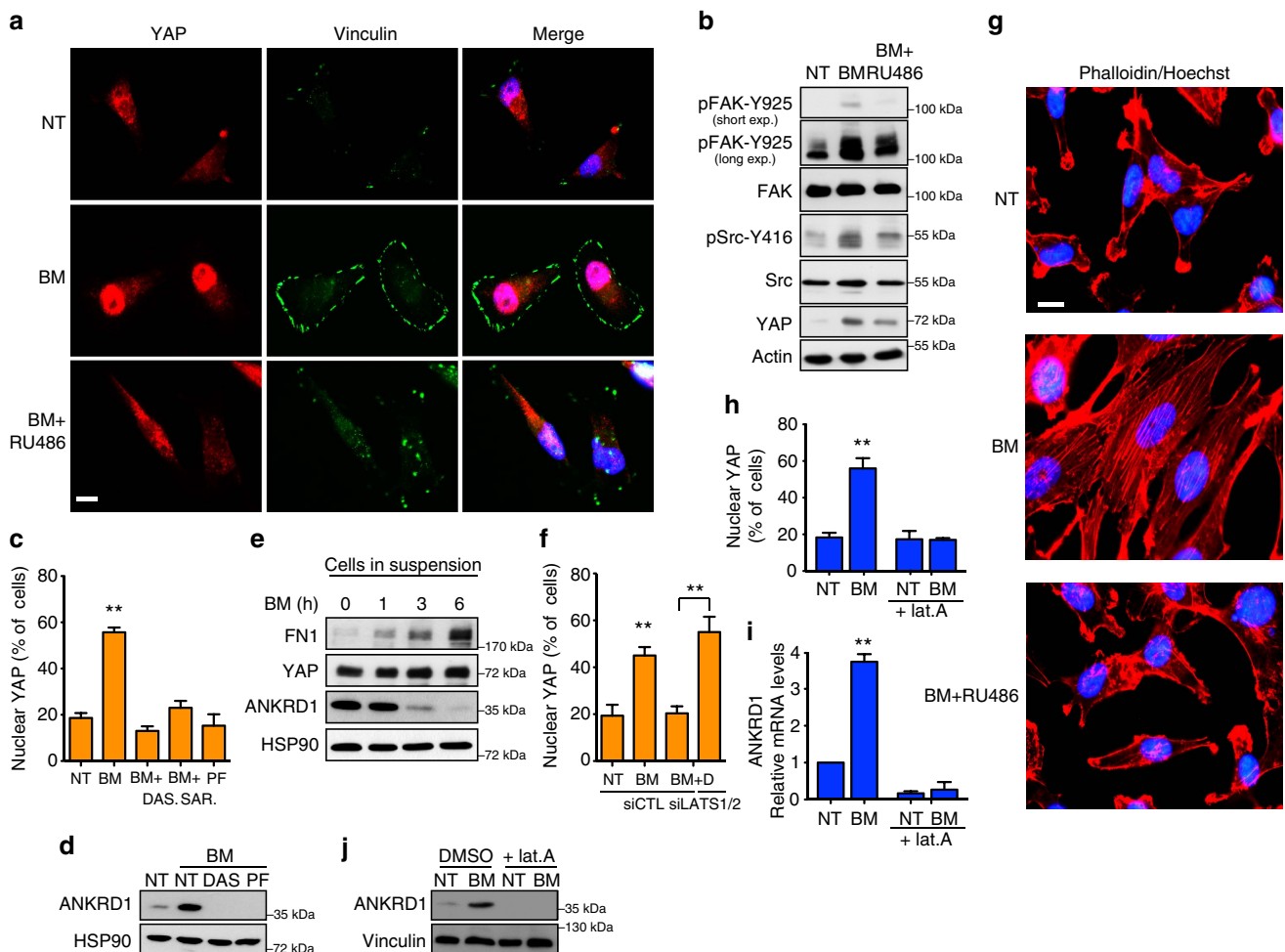

**Figure 5 | Glucocorticoids activate YAP via FAK/Src-dependent actin cytoskeleton remodelling.** (**a**) MDA-MB-231 cells were grown in serum-free medium and treated with betamethasone (BM) 1 μM alone or in combination with RU486 1 μM for 24 h. Representative images of immunofluorescence are shown. Experiment repeated three times. Scale bars, 15 μm. (**b**) MDA-MB-231 cells were treated as in **a**. Representative blots are shown. Experiment repeated three times. (**c**) Quantification of MDA-MB-231 cells with nuclear YAP by immunofluorescence. Cells were grown in serum-free medium in presence of BM 1 μM alone or in combination with dasatinib (DAS) 0.1 μM or saracatinib (SAR) 0.1 μM or PF573228 (PF) 5 μM for 24 h. Error bars represent mean ± s.d., from n = 3 biological replicates. (**d**) MDA-MB-231 cells were grown in serum-free medium in presence of BM 1 μM alone or in combination with DAS 0.1 μM or PF 5 μM for 24 h. Representative blots are shown. Experiment repeated three times. (**e**) MDA-MB-231 cells were trypsinized and then maintained in suspension and treated with BM 1 μM for indicated times. Representative blots are shown. (**f**) Quantification of MDA-MB-231 cells with nuclear YAP by immunofluorescence. Cells were transfected with control siRNA (siCTL) or a combination of siRNA targeting LATS1 and 2 (siLATS1/2). The day after cells were grown in serum-free medium in presence of BM 1 μM alone or in combination with DAS 0.1 μM for 24 h. Error bars represent mean ± s.d., from n = 3 biological replicates. (**g**) Representative images of immunofluorescence. MDA-MB-231 cells were grown in serum-free medium and treated with BM 1 μM alone or in combination with RU486 1 μM for 24 h. Experiment repeated three times. Scale bars, 15 μm. (**h**) Quantification of MDA-MB-231 cells with nuclear YAP by immunofluorescence. Cells were grown in serum-free medium in presence of BM 1 μM alone or in combination with latrunculin A (lat.A) 0.5 μM for 24 h. Error bars represent mean ± s.d., from n = 3 biological replicates. (**i**) Quantitative PCR with reverse transcription analysis of MDA-MB-231 cells grown in serum-free medium in presence of BM 1 μM alone or in combination with lat.A 0.5 μM for 24 h. Error bars represent mean ± s.d., from n = 3 biological replicates. (**j**) MDA-MB-231 cells were treated as in **g**. Representative blots are shown. Experiment repeated three times. *$P < 0.05$, **$P < 0.01$; two-tailed Student's t-test is used throughout.

**YAP is required for GC-induced stem cell traits.** Expression of YAP in breast basal cells triggers CSC formation and maintenance via acquisition of stem cell traits[34]. On the basis of our results, we hypothesized that the endocrine control exerted by GCs on YAP might have an impact in sustaining the self-renewal potential of breast CSCs[35]. To assess whether the GR–YAP axis plays a causal role in defining CSC traits, we tested the effect of BM on the self-renewal potential of mammary epithelial cells from a panel of different basal breast cancer cell lines by testing their capacity to form and propagate mammospheres *in vitro*[36–38]. Strikingly, a 5 days

GC pre-treatment[39] of MII, MDA-MB-231, BT-549 and SUM159 cells grown in 2-day culture conditions, significantly increased the efficiency of secondary mammosphere formation (Fig. 6a,b), suggesting that activation of GR signalling potentiates CSCs' self-renewal. In line, the treatment increased the percentage of CD44+/CD24− cells, a population of cells with high self-renewal potential, in a GR-dependent manner (Supplementary Fig. 6a)[34]. To further confirm these data, we investigated the involvement of GR signalling in CSC traits in human breast tumours. To this aim, we retrieved two signatures of GR activation from two independent published

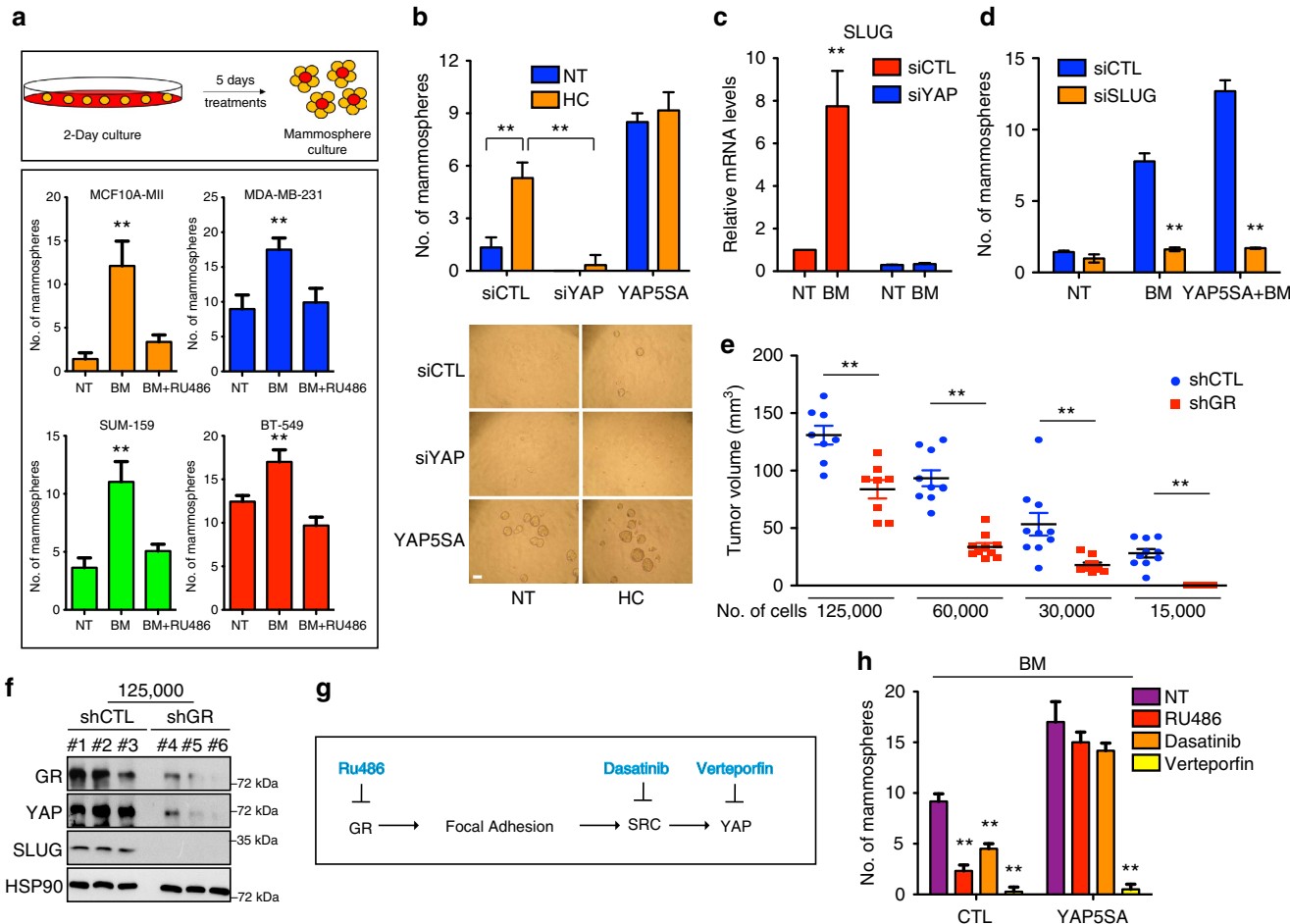

**Figure 6 | YAP is required for glucocorticoids-induced stem cells traits in breast cancer cells.** (**a**) Upper panel: schematic representation of the experiment. Cells were grown in 2-day culture and treated with indicated compounds for 5 days, then cells were cultured in mammosphere conditions. Lower panel: number of secondary mammospheres generated from the indicated breast cancer cell lines treated with vehicle (NT) or betamethasone 1 µM (BM) alone or in combination with RU486 1 µM. Error bars represent mean ± s.d., from $n = 3$ biological replicates. (**b**) Upper panel: number of secondary mammospheres generated from MII cells overexpressing control vector (CTL) or YAP5SA, transfected with control (siCTL) or YAP (siYAP) siRNA and treated as indicated. HC is hydrocortisone 1 µM. Lower panel: representative images of mammospheres. Error bars represent mean ± s.d., from $n = 3$ biological replicates. siYAP sequence is siYAP#1. (**c**) Quantitative PCR with reverse transcription analysis of MDA-MB-231 transfected with indicated siRNA for 48 h and treated with 1 µM BM alone or in combination with RU486 1 µM for 24 h. siCTL is control siRNA. Error bars represent mean ± s.d., from $n = 3$ biological replicates. siYAP sequence is siYAP#1. (**d**) Number of secondary mammospheres generated from MII cells overexpressing control vector (CTL) or YAP5SA, transfected with indicated siRNA and treated as in **a**. Error bars represent mean ± s.d., from $n = 3$ biological replicates. (**e**) Tumour volumes 21 days after injection of indicated number of MDA-MB-231-shCTL or MDA-MB-231-shGR cell into mammary fat pad of mice. Error bars represent mean ± s.e.m. (**f**) Lysates of tumours from 125,000 MDA-MB-231-shCTL or MDA-MB-231-shGR cells injected in mice were immunoblotted with the indicated antibodies. The numbers are mice identificative numbers. (**g**) Schematic representation summarizing drugs and their targets. (**h**) Number of secondary mammospheres from MII cells overexpressing control vector (CTL) or YAP5SA. Cells were treated as in **a** with indicated compounds. Error bars represent mean ± s.d., from $n = 3$ biological replicates. $*P < 0.05$, $**P < 0.01$; two-tailed Student's $t$-test is used throughout.

datasets obtained from GC-stimulated genes in MDA-MB-231 and A549 cells[25,40] and found that GR transcriptional activity is associated with molecular signatures of breast stem cells in a metadata set of 3,661 primary human breast cancers (Supplementary Fig. 6e)[41,42].

To study the impact of YAP in GC-induced self-renewal, we performed mammosphere assays on YAP knockdown and HC treatment. Of note, YAP ablation completely prevented the increase of mammospheres number induced by HC (Fig. 6b; Supplementary Fig. 6b). Moreover, to formally prove that the effect of GCs on mammosphere formation and expansion was dependent on YAP nuclear activation, we stably transduced MII cells with the constitutively active form of YAP (YAP-5SA) and found that the transduced cells were completely refractory to GC treatment being able to form mammospheres in the absence of HC in the medium (Fig. 6b).

To gain insights into the mechanism of CSC expansion by GCs via YAP activation we analysed the list of genes induced by GC treatment (Supplementary Table 1) and identified Slug (also known as SNAI2), a crucial determinant of breast CSCs traits[43], among the GC-induced genes. Of note, Slug is a known YAP target gene[17,44,45], therefore we hypothesized that GC could sustain the expansion of CSCs by increasing the expression of Slug in a YAP-dependent manner. To test this, the expression of Slug was analysed in cells treated with BM. As shown in Fig. 6c, BM induced a strong increase of Slug mRNA levels, while YAP depletion completely prevented this effect. In line with this, Slug knockdown efficiently inhibited

the positive effects of BM on mammosphere formation (Fig. 6d; Supplementary Fig. 3f), suggesting that downstream of YAP, Slug acts as an effector of the GC signalling to establish CSCs traits.

Next, we wanted to test whether GR affects the ability of tumour progenitor cells to seed tumours *in vivo* by serial dilution transplantation experiment of MDA-MB-231-shCTL and MDA-MB-231-shGR cells in mice. Of note, GR depletion reduced the tumour size and the frequency of tumour engraftment (Fig. 6e). As expected, YAP and Slug protein levels were reduced in tumours from MDA-MB-231 cells depleted of GR (Fig. 6f). These results are consistent with the *in vitro* results described above and demonstrate that GR signalling is required for the maintenance of tumour-initiating cells.

Finally, we assessed whether inhibition of the entire GR/YAP axis might represent a pharmacological strategy to specifically target CSCs in breast cancer. This was accomplished using drugs acting at three different steps (Fig. 6g): GR inactivation by means of RU486, Src inactivation by dasatinib and YAP inactivation by verteporfin, which inhibits the physical YAP–TEAD interaction[46]. Interestingly, all these inhibitors dramatically interfered with the BM-induced self-renewal of CSCs in MDA-MB-231 and MII cells (Fig. 6h; Supplementary Fig. 6c). Similar results were obtained on FN1 knockdown (Supplementary Fig. 6d). Overexpression of nuclear YAP rescued the effect of RU486 and dasatinib but not of verteporfin, consistently with our results indicating that GR and Src act upstream of YAP (Fig. 6h).

**GR-dependent YAP activation is involved in chemoresistance**. To assess whether GR signalling correlates with YAP activation in human breast cancer, we stratified patients from a meta-data set of primary human breast tumours into groups displaying high or low GR pathway activation and assessed the level of YAP activity using a published YAP signature[16]. As shown in Fig. 7a, patients classified as having high GR activation also showed high YAP activity, thus confirming our *in vitro* results (Fig. 7a; Supplementary Fig. 6f).

In estrogen receptor (ER)-negative breast cancers, including triple-negative breast cancer and in prostate cancer, expression levels of GR correlate with bad prognosis of chemotherapy-treated patients, suggesting a role for GR in tumour aggressiveness and resistance to drug treatment[25,47,48]. However, although this evidence has been well documented, the mechanisms underlying the GR-associated chemoresistance in breast cancer are largely unknown. In addition to self-renewal, resistance to standard chemotherapy (for example, taxanes treatment) is another well-established feature of CSCs[49]. On the basis of our results, we hypothesized that the GC-induced chemoresistance could be mediated by YAP that acts downstream of GR to fuel the expansion of drug-resistant CSCs. To test this hypothesis, we treated MDA-MB-231 cells with paclitaxel (PX) for 48 h and assessed cell death by monitoring the cleavage of PARP-85, a marker of apoptosis. As expected, PX elicited apoptosis while co-treatment with BM efficiently prevented cell death in a GR-dependent manner (Supplementary Fig. 6g). To demonstrate the involvement of YAP in BM-induced cell survival we knocked down YAP in PX-treated MDA-MB-231 cells and found that YAP silencing rescued drug-induced cell death in BM-treated cells (Fig. 7b). This evidence prompted us to investigate the consequences of the GR–YAP axis abrogation on CSCs expansion during drug treatment. To this aim, we elicited mammosphere formation from MII and MDA-MB-231 cells treated with PX. As expected, PX promoted a slight increase of mammosphere formation efficiency due to the selective expansion of chemotherapy-resistant CSCs[50] (Fig. 7c).

However, the pharmacological inhibition of the GR pathway by RU486 co-treatment completely blunted this effect, suggesting that GR is responsible for the expansion of CSCs during PX treatment (Fig. 7c). Moreover, YAP-5SA overexpression completely rescued CSC expansion in the presence of PX and RU486, thus confirming the role of YAP as an effector of the GR in inducing CSC self-renewal and chemoresistance (Fig. 7c,e). In line with these results, we stratified patients based on the activation level of GR and we found that high activity of GR was associated with shorter overall survival in basal-like breast tumours (Fig. 7d).

## Discussion

In this report, we demonstrate that YAP is a sensor of GR signalling in breast cancer cells. Mechanistically, we found that GR signalling affects the mechanical properties of the tumour microenvironment, ultimately promoting YAP nuclear accumulation and activation. On hormonal stimulation, GR induces expression and extracellular deposition of fibronectin, leading to increased number of focal adhesions and cell spreading. These events, in turn, activate Src and induce actin cytoskeleton rearrangements that promote YAP activation. The GC–GR–YAP axis is required for and/or enhances self-renewal of breast CSCs and YAP is responsible for the GC-induced resistance to chemotherapy drugs (Fig. 7c).

Alterations of the mechanical properties of the ECM, that is, stiffness and elasticity, can be induced by cancer cells themselves or by stromal cells and represent mechanical inputs that profoundly affect crucial biological aspects of tumour development, such as cell proliferation, differentiation and apoptosis[29]. Breast tumours are typically stiffer than the normal tissue and this feature is commonly used as a diagnostic tool to screen patients for early lesions[51]. The mechanisms underlying how cells sense or alter the mechanical properties of the microenvironment in response to certain stimuli remain largely unknown, particularly in the context of epithelial cells. In this regard, our results establish a direct relationship between the endocrine system and the mechanical features of the ECM, opening a scenario in which aberrant endocrine activation of GR has a relevant impact on tumour microenvironment and, as a consequence, on cancer aggressiveness.

Cells can sense mechanical signals from ECM through various physical contacts, among which integrin-based cell–matrix focal adhesions. Our data indicate that GR stimulation activates YAP through a hormonal and cell-autonomous manner and involves intracellular signalling pathways triggered by the cell–ECM interplay. On sensing forces, cells react by increasing tension within the intracellular actin cytoskeleton. This mainly occurs through the activation of two families of proteins: the Src kinases and the RhoGTPases. Signalling pathways activated by these enzymes have been found to mediate the effects of ECM rigidity on YAP[18,27,32,52]. Although RhoA activation on GC treatment has been previously reported[53], in this study the main route through which GR signalling acts to activate YAP relies on integrin/Src activation and F-actin polymerization. As the signalling pathways activated by Src kinases and RhoGTPases are connected by extensive crosstalk, further investigations are needed to clarify the exact role of RhoGTPases in this context.

Interestingly, recent work reported that, through YAP/TAZ activation, ECM stiffness profoundly affects the sensitivity of breast cancer cells to therapeutics[54]. In the clinic, GCs are frequently prescribed to cancer patients to alleviate the acute toxicity of chemotherapy and to protect healthy tissue (for example, bone marrow) against its long-term effects[55].

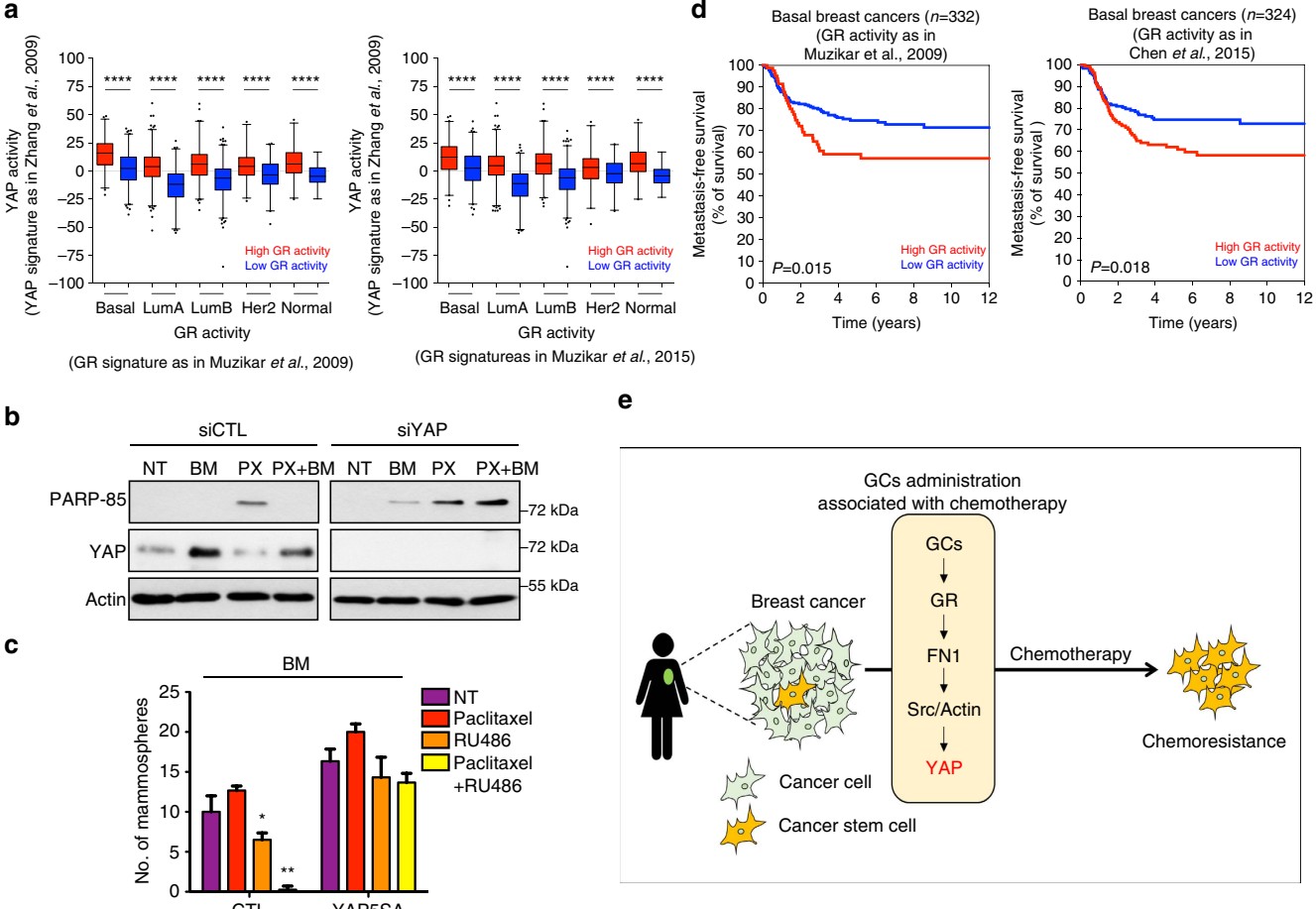

**Figure 7 | Glucocorticoid receptor activation correlates with YAP activity in breast cancer and is involved in chemoresistance.** (**a**) Primary human breast cancers of the metadata set were stratified according to high or low GR activity signature (left panel[40]; right panel[25]) and then, the levels of the YAP activity signature score were determined in the intrinsic molecular subtypes (PAM50). YAP activity is significantly higher in breast cancers with higher levels of the GR activity signature, as visualized by the box plot. Signature scores have been obtained, summarizing the standardized expression levels of signature genes into a combined score with zero mean[70]. The values shown in graphs are thus adimensional. The bottom and top of the box are the first and third quartiles, and the band inside the box is the median; whiskers represent 1st and 99th percentiles; values that are lower and greater are shown as circles (****$P < 0.0001$, two-tailed Student's $t$-test). (**b**) MDA-MB-231 cells transfected with control (siCTL) or YAP (siYAP) siRNA and treated with vehicle (NT) or betamethasone 1 μM (BM) and paclitaxel 0,1 μM (PX) alone or in combination for 48 h. Representative blots are shown. Experiment repeated three times. siYAP sequence is siYAP#1. (**c**) Number of secondary mammospheres generated from MII cells overexpressing control vector (CTL) or YAP5SA and treated as in a, with indicated compounds. Error bars represent mean ± s.d., from $n = 3$ biological replicates. (**d**) Kaplan–Meier analysis representing the probability of disease-specific survival in basal breast cancer from the metadata set stratified according to high or low GR signature score (left panel[40]; right panel[25]). The log-rank test $P$ value reflects the significance of the association between high levels of the GR signature score and shorter survival in GR signature high as compared with GR signature low patients ($P < 0.05$). (**e**) Proposed working model. *$P < 0.05$, **$P < 0.01$; two-tailed Student's $t$-test is used throughout.

However, mounting clinical evidence has suggested that GR activation may induce therapy resistance and is associated with bad prognosis in solid tumours (for example,, basal-like breast cancer and prostate cancer)[25,47,48]. Data presented in this work not only fuel the concern raised over the use of GCs as supportive co-medication in anti-cancer treatments, but also provide novel mechanistic explanations of how GCs lead to chemoresistance and tumour relapse by promoting the unscheduled expansion of CSCs, a sub-population of cancer cells that is intrinsically resistant to therapy and highly metastatic[49].

In our experimental set-up, GCs activate YAP at doses that are comparable to GCs plasma concentrations detected in cancer patients receiving these hormones as palliative therapy[56]. This supports the notion that the intrinsic ability of GCs to aberrantly activate YAP and sustain CSCs may contribute to GCs effects on chemotherapy response.

In line with our observations, it has been found that RU486 treatment in mice sensitizes tumours from MDA-MB-231 xenografts to PX treatment[57]. On the basis of our results, it is conceivable that YAP inhibition and exhaustion of CSCs could mediate this effect of RU486. Of note, an ongoing Phase 1 clinical trial is testing the safety and tolerability of RU486 in combination with nab-PX for advanced ER-negative, PR-negative and HER2-negative breast cancer (NCT01493310).

Our findings, as also reported in other studies[16,58,59], demonstrate the existence of a functional difference of the Hippo pathway effectors YAP and TAZ. The major structural differences between YAP and TAZ and their different repertoire of post-translational modification and interacting proteins[60] might explain the divergent response of these two proteins to GR stimulation.

GCs are steroid hormones regulated by the circadian rhythm and in response to several types of stress. The influence of acute and chronic stress factors on the development and progression of cancer has been a longstanding hypothesis since ancient times. Epidemiological and clinical studies over the past 30 years have provided evidence for a link between dysregulation of neuroendocrine hormones, in particular catecholamines and cortisol, and cancer progression[58]. The unpredicted role of GCs on YAP activation unveiled by this work suggests that a direct link between stress hormones and oncogenes does exist and that, in some cases, oncogenes may be activated in an endocrine manner.

In spite of their pro-oncogenic role in some solid cancers, GCs have well-documented pro-apoptotic activity in cells from haematological tumours and are considered as a first line of defence in the treatment of different leukaemias[59]. Interestingly, a recent study has reported YAP as an activator of ABL1/p73-mediated apoptosis in the context of haematological malignancies[60,61]. This evidence and our results may suggest other possible connections between GCs and YAP via different mechanisms.

In light of our findings, in fact, it is conceivable that YAP activity could be involved in diverse GC-induced biological responses, in both physiological and pathological conditions. The metabolic syndrome (MetS), for example, might provide another relevant pathological context where this relation is worthy to be investigated. MetS is a clustering of condition predisposing to cardiovascular disease and cancer, among other co-morbidities[62]. Mounting clinical data and animal genetic studies have strongly associated aberrant GC signalling with diverse features of MetS[63].

From a different point of view, the identification of GCs as Hippo pathway regulators opens the doors to the exploration of the potential role both of GCs in organ size control during development and of YAP as part of the cellular physiological response to GCs released under stress or circadian rhythm. The functional role of the endocrine system in YAP activation thus provides a novel conceptual framework to better understand YAP/Hippo pathway functions and regulatory mechanisms under a number of different signals and conditions.

## Methods

**Reagents and plasmids.** The library of FDA-approved drugs (Screen-Well FDA-Approved Drug Library, 640 chemical compounds dissolved at 10 mM in dimethylsulphoxide (DMSO)) was obtained from Enzo Life Sciences (Enzo Life Sciences Inc.,Plymouth Meeting, PA, USA).

The following compounds were purchased from Sigma Aldrich: HC (H0888), verteporfin (SML0534), PX (T7191), RU486 (Mifepristone; M8046), BM (B7005), PF5573228 (PZ0117). RGD is fibronectin tetrapeptide (H-Arg-Gly-Asp-Ser-OH) from Santa Cruz (sc-202156).

Latrunculin A (sc-202691) was from Sigma, saracatinib (S1006) and dasatinib (S1021) from Selleck.

8XGTII-Luc and the retroviral construct coding for Flag-YAP-5SA were previously described[16]. Lentiviral constructs coding for short hairpin NR3C1 (GR) was from Sigma (Clone ID:NM_000176.2-6190s21c1).

**Cell lines and isolation and purification of mammary epithelial cells.** Cell lines were obtained from American Type Culture Collection (ATCC) or from other laboratories cooperating on the project. MDA-MB-231 and BT-549 were cultured in DMEM supplemented with 10% FBS and antibiotics. MCF10A MII cells were previously described[16]. Cells were obtained from ATCC or from other laboratories cooperating on the project. Cells were subjected to STR genotyping with PowerPlex 18D System and confirmed in their identity comparing the results to reference cell databases (DMSZ, ATCC and JCRB databases). All cell lines have been tested for mycoplasma contamination.

Mammary glands from 8 to 12-week-old virgin female mice were enzymatically digested and single-cell suspensions of purified mammary epithelial cells were obtained, as described[64,65]. Briefly, mammary glands from 8 to 12-week-old virgin female mice were digested for 1–2 h at 37 °C in Epi- Cult-B medium (StemCell Technologies Inc, Vancouver, Canada) with 600 U ml$^{-1}$ collagenase

(Sigma-Aldrich, St Louis, MO, USA) and 200 U ml$^{-1}$ hyaluronidase (Sigma). After lysis of the red blood cells with NH$_4$Cl, the remaining cells were washed with PBS/0.02% w/v EDTA to allow cell–cell contacts begin to break down. Cells were then dissociated with 2 ml trypsin 0.25%w/v, 0.2% w/v EDTA for 2 min by gentle pipetting, then incubated in 5 mg ml$^{-1}$ Dispase II (Sigma) plus 1 µg ml$^{-1}$ DNase I (Sigma) for 5 min followed by filtration through a 40 µM cell strainer (BD Falcon, San Jose, CA, USA). Mammary epithelial cells were then purified using the EasySep Mouse Mammary Stem Cell Enrichment Kit (StemCell Technologies Inc).

**High-content screening.** For the screening experiments, MDA-MB-231 cells ($3.0 \times 10^3$ per well) were seeded on black clear-bottom 384-well plates (PerkinElmer). Twenty-four hours later, the FDA-approved drugs were transferred robotically from library stock plates (0.1 and 1 mM in DMSO) to the plates containing the cells; controls were added to columns 1, 2, 23 and 24 of each plate. Cells were fixed at 48 h after plating, that is, 24 h after addition of drugs, and processed immediately for immunofluorescence. Briefly, cells were fixed with 4% paraformaldehyde for 15 min, permeabilized with 0.1% Triton X-100 in phosphate-buffered saline (PBS) solution for 10 min, followed by 30 min blocking in 3% FBS. Cells were then incubated with a mouse antibody against YAP (Santa Cruz Biotechnology) diluted in blocking solution for 1 h. Cells were further washed with PBS and incubated for 1 h with a secondary antibody conjugated to Alexa Fluor-594 (Life Technologies) and stained with Hoechst 33342 (Life Technologies).

Image acquisition was performed using an ImageXpress Micro automated high-content screening fluorescence microscope (Molecular Devices) at a × 10 magnification; a total of nine images were acquired per wavelength, well and replicate, corresponding to ca. 4,500 cells analysed per experimental condition and replicate. Image analysis to quantify the intensity of YAP and to identify cells presenting predominantly nuclear YAP localization was performed using the 'Multi-Wavelength Cell Scoring' and 'Multi-Wavelength Translocation' application modules implemented in MetaXpress software (Molecular Devices).

Screening was performed in duplicate, at two drug concentrations (1 and 10 µM); final concentration of DMSO in the culture medium was 1% (v/v) for all experimental conditions. The screening was performed at the ICGEB High-Throughput Screening Facility (http://www.icgeb.org/high-throughput-screening.html).

**Transfections.** siRNA transfections were done with Lipofectamine RNAi-MAX (Life technologies) in antibiotics-free medium according to manufacturer instructions. siRNAs were previously described[18] and sequences are shown in Supplementary Table 2.

Negative control siRNA was: AllStars negative control siRNA Qiagen 1027281.

DNA transfections were done with Lipofectamine LTX & Plus Reagent (Invitrogen) or Lipofectamine RNAiMAX (Life Technologies) according to manufacturer instructions. Lentiviral particles were prepared by transiently transfecting HEK293T cells with lentiviral vectors together with packaging vectors (pMD2-VSVG and psPAX2) using the standard calcium phosphate method. Retroviruses were made by calcium phosphate transfection of HEK293-GP packaging cells with the appropriate plasmids in combination with pMD2ENV coding for envelope proteins, and collected 48 h later. Infected cells were selected with puromycin 2 mg ml$^{-1}$.

**Luciferase assay.** Luciferase assays were performed in MDA-MB-231 cells and in BT-549 cells with the established YAP/TAZ-responsive reporter 8XGTII-Luc[18]. Luciferase reporters (300 ng cm$^{-2}$) were transfected together with CMV-Renilla (30 ng cm$^{-2}$) to normalize for transfection efficiency. For luciferase assays in siRNA-transfected cells, cells were first transfected with the indicated siRNAs and, after 24 h, washed from transfection media, transfected with plasmid DNA and collected 24 h later.

**Soft agar assay.** MDA-MB-231 cells in complete growth medium with 0.3% agar were layered onto 1% agar beds in six-well plates; complete medium with 1 µM BM was added on top of cells and was replaced with fresh medium twice a week for 15 days. Colonies larger than 100 µm in diameter were counted as positive for growth.

**Mammosphere assay.** Mammosphere assays were performed as previously described[38]. In brief, cells were treated with the indicated drugs for 5 days in 2-day culture. Then, cells were trypsinized, stained with trypan blue to check cell viability and plated as single-cell suspension on ultra-low attachment plates (Corning) in mammospheres growing medium[36,38]. When spheres reached > 100 µm size, spheres were counted, harvested with p1000, washed with PBS, pelleted and dissociated with trypsin. Single cells where stained with trypan blue, counted and reseeded for a second round of mammospheres formation.

**Fluorescence-activated cell sorting.** Fluorescence-activated cell sorting assays were performed as previously described[38]. Cells were detached from plates with trypsin, incubated in running buffer (PBS 1×, BSA 0.5% and EDTA 5 mM) with anti-human PE conjugated CD44 and anti-human fluorescein isothiocyanate-conjuugated CD24 (BD Biosciences) and finally analysed with ARIA II cell sorter (Beckton Dickinson).

**Antibodies.** Antibodies used for WB and immunofluorescence: anti-YAP is sc101199 (1:1,000, Santa Cruz Biotechnology), anti- GR is BK3660S (1:1,000, Cell Signaling), anti-actin is C11 (1:2,000, Sigma), anti-ANKRD1 is 11427-1-AP (1:1,000, Proteintech (DBA), anti-pYAP (Ser127) is 4911S (1:1,000, Cell Signaling), anti-WWTR1 is HPA007415 (1:1,000, Sigma), phalloidin is A12379 (1:250, Alexa Fluor), anti-PARP-85 is TB273 (1:500, Promega), anti-vinculin is V4505 (1:4,000, Sigma), anti-LATS1 is ab70562 (1:500, Abcam), phospho-LATS1 (Thr1079) is BK8654S (1:500, Cell Signaling), anti-FAK (C-20) is sc-558 (1:1,000, Santa Cruz), anti-FAK (phospho Y397) is ab81298 (1:1,000, Abcam), anti-Src is BK2110S and anti-Phospho-Src (Tyr416) is BK2101S from Cell Signaling (1:1,000), anti-FN1 is GTX112794 (1:1,000, GeneTex), anti-Slug is C19G7 (1:500, Cell Signaling), anti-Mst1 is 3682S (1:500, Cell Signaling) and anti-phospho-Mst1/2 (T183/T180) is 3681S (1:500, Cell Signaling).

**Quantitative real-time PCR.** Cells were collected in Qiazol lysis reagent (Qiagen) for total RNA extraction, and contaminant DNA was removed by DNase treatment. Quantitative PCR with reverse transcription analyses were carried out on retrotranscribed complementary DNAs with Quantitect reverse transcription kit (Qiagen) and analysed with Biorad CFX Manager software. Experiments were performed at least three times, with duplicate replicates. Expression levels are always given relative to histone H3. PCR oligo sequences for human samples are shown in Supplementary Table 3.

**Immunofluorescence and WB.** Immunofluorescence staining was performed as previously described[11]. Briefly, cells were fixed in 4% paraformaldehyde for 10 min, washed in PBS, permeabilized with Triton 0.1% for 10 min and blocked in PBS FBS 3% for 30 min. Antigen recognition was done by incubating primary antibody for 1 h at 37 °C and with goat anti-mouse Alexa Fluor 568 (Life Technologies) as secondary antibody for 30 min a 37 °C. Nuclei were counterstained with Hoechst 33342 (Life Technologies). For extracellular fibronectin staining, cells were plated onto glass coverslips without fibronectin coating and immunofluorescence was performed without permeabilization.

WB analysis was performed as previously described[11]. Uncropped blots are shown in Supplementary Fig. 7.

**Chromatin immunoprecipitation.** Following 6 h of treatment with BM 1 µM, MDA-MB-231cells were crosslinked for 15 min with 1% formaldehyde, neutralized with 125 mM glycine pH 2.5 and washed in PBS. Cells were scraped and centrifuged at 1700 g for 10 min at 4 °C. Pellets were resuspended in 10 ml Chro-IP Lysis Buffer (50 mM Hepes-KOH at pH 8; 1 mM EDTA; 0.5 mM EGTA; 140 mM NaCl; 10% glycerol; 0.5% NP-40; 0.25% Triton X-100; protease inhibitor cocktail (Sigma), 1 mM PMSF, 5 mM NaF) for 10 min at 4 °C. The crude nuclei were collected by centrifugation (1300g for 5 min at 4 °C), resuspended in 10 ml wash buffer (10 mM Tris-HCl at pH 8; 1 mM EDTA; 0.5 mM EGTA; 200 mM NaCl; protease inhibitor cocktail (Sigma), 1 mM PMSF, 5 mM NaF)for 5 min at 4 °C. Washed nuclei were centrifuged as described earlier and resuspended in 2 ml of RIPA 140 mM (10 mM Tris-HCl at pH 8; 1 mM EDTA; 0.5 mM EGTA; 140 mM NaCl; 1% Triton X-100; 0.1% Na-deoxycholate; 0.1% SDS; protease inhibitor cocktail (Sigma), 1 mM PMSF, 5 mM NaF). Samples were sonicated (power setting 5) with a Misonix Microson in 10" bursts followed by 50" of cooling on ice for a total sonication time of 2 min per sample. Chromatin was precleared for 1 h at 4 °C with protein A/G PLUS-Agarose (Santa Cruz Biotechnologies) and subsequently immunoprecipitated overnight at 4 °C with 2 µg of anti-GR (BK3660S, Cell Signaling); normal Rabbit IgG (sc-2027 Santa Cruz) were used as negative control. DNA protein complexes were recovered with protein A/G PLUS-Agarose and washed sequentially with RIPA 140 mM buffer, RIPA 250 mM buffer (20 mM Tris-HCl pH 7.5, 250 mM NaCl, 1 mM EDTA, 0.5% NP-40, 0.5% Na-deoxycholate and 0.1% SDS), LiCl solution (10 mM Tris-HCl pH 8.0, 1 mM EDTA, 250 mM LiCl, 0.5% NP-40 and 0.5% Na-Deoxycholate) and TE. RNase treatment was performed in TE for 30 min at 37 °C and to reverse cross-linking samples were treated overnight at 68 °C adding an equal volume of proteinase K solution (200 mM NaCl, 1% SDS and 0.3 mg ml⁻¹ proteinase K). In parallel, inputs were treated in the same way. After phenol/chloroform extraction and ethanol precipitation samples were resuspended in $H_2O$.

Real-time PCR was performed by using iTaq Universal SYBR Green Supermix (BIORAD). Primer sequences were retrieved from ENCODE data and are shown in the Methods section (Quantitative real-time PCR). Promoter occupancy was calculated as the fold increase of normalized immunoprecipitated chromatin over the control IgG with the $2^{-\Delta\Delta Ct}$ method.

**Biostatistical analysis.** RNA-Seq data of MDA-MB-231 cells treated with DM were retrieved from GSE56022 (ref. 25). We defined GR target genes as genes with a fold change >1.3 and an increase in expression >1 RPKM (reads per kilobase per million mapped reads) in at least one replicate of cells treated with 100 nM DM for 4 h as compared with cells treated with vehicle. Enrichment analysis was conducted using David functional annotation. ChIP-Seq data for GR in A549 cells treated with 100 nM DM were retrieved from ENCODE[66].

The metadaset of breast cancer gene expression profiles has been created starting from a collection of 4,640 samples from 27 major data sets, comprising microarray data of breast cancer samples annotated with histological tumour grade and clinical outcome[17,67]. The collection was normalized and annotated with clinical information as described in ref. 67. This resulted in a compendium comprising 3,661 unique samples from 25 independent cohorts. Intrinsic molecular subtypes were assigned using the *intrinsic.cluster.predict* function of *genefu* R package[68] using the '50 intrinsic gene list' as proposed by Parker and colleagues (PAM50; ref. 69).

To identify two groups of tumours with either high or low GR activity signature, we used the classifier described in ref. 70. Tumours were classified as GR signature low if the combined score was lower than the first quartile of the score distribution and as GR signature high if the combined score was higher than the third quartile of the score distribution. This classification was applied to expression values of the metadata set.

GR activity signatures were derived from GCs-induced genes in MDA-MB-231 and A549 cells[25,40]. Briefly, to derive the GR signature from the data of Chen et al.[25] we started from the list of GR target genes in MDA-MB-231 cells treated with DM and further refined it selecting only those genes overexpressed in cells treated with DM (fold change >2 and an RPKM difference ≥2), but not in cells treated with Compound A (fold change ≤2) in the comparison with cells treated with vehicle at 4 h (ref. 25). To derived the GR signature in A549 lung adenocarcinoma epithelial cells[40], we downloaded the raw data of GSE17307 from Gene Expression Omnibus and converted probe level signals to expression values using robust multi-array average procedure RMA[71] of Bioconductor *affy* package. Differentially expressed genes were identified using Significance Analysis of Microarray algorithm coded in the *samr* R package[72]. In SAM, we estimated the percentage of false-positive predictions (that is, false discovery rate, FDR) with 100 permutations and defined GC-induced genes selecting, at a confidence level of 95% (FDR ≤ 5%), those probes with a fold change ≥2 in cells treated with DM (for 6 h) and with a negative fold change in cells treated with DM and GR antagonist mifepristone. The full list of genes enlisted in the GR activity signatures is provided in Supplementary Table 1.

Average stem cell and Yap signature expression has been calculated as the standardized average expression of all signature genes in sample subgroups (for example, GR activity high/low). As stem cell signatures we used the lists of genes of Shipitsin et al.[41] and of Farmer et al.[42] The YAP activity signature is composed of a selection of genes that have been found activated by YAP overexpression in human mammary cells (MCF10A; ref. 73).

To evaluate the prognostic value of the GR signature, we estimated, using the Kaplan–Meier method, the probabilities that patients would remain free of metastatic events. To confirm these findings, the Kaplan–Meier curves were compared using the log-rank (Mantel–Cox) test. P values were calculated according to the standard normal asymptotic distribution. Survival analysis was performed in GraphPad Prism.

The sample size was chosen to include at least three biological replicates with two technical replicates each. Experiments for which we showed representative images were performed successfully at least 3 independent times. No samples or animal were excluded from the analysis.

No statistical method was used to predetermine the sample size for animal studies. Standard laboratory practice randomization procedure was used for cell line groups and animals of the same age and sex. The investigators were not blinded to allocation during experiments and outcome assessment. All P values were determined using two-tailed t-tests and statistical significance was set at P = 0.05. The variance was similar between groups that we compared.

**Mice and animal care.** Eight-week-old C57BL/6 female mice were maintained in a specific pathogen free animal facility. Mice received a single intraperitoneal bolus of DM sulfate (20 mg kg⁻¹) at time 0, to be sacrified 16 h later for mammary gland collection. Procedures involving animals and their care were in conformity with institutional guidelines (D.L. 116/92 and 26/2014, and subsequent implementing circulars) and all experimental protocols were approved by the ethical Committee of the University of Padua (CEASA). For limiting dilution experiments serial dilutions of MDA-MB-231-shCTL or MDA-MB-231-shGR cells ranging from 125,000 to 15,000 cells were resuspended in 100 µl of DMEM, and injected into the mammary fat pad of previously anaesthetized (1–3% isoflurane, Merial Italia) NSG female mice. Tumour growth at the injection site was monitored by caliper measurements and tumour volume was calculated using the formula: Tumour volume $(mm^3) = D \times d^2/2$, where D and d are the longest and the shortest diameters, respectively.

**Data availability.** The authors declare that the main data supporting the findings of this study are available within the article and its Supplementary Information Files. Extra data are available from the corresponding author on request.

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

## Acknowledgements

We thank A. Testa for discussions and proofreading the manuscript and A. Zambelli for discussions. We acknowledge G. Pastore for technical support. We thank S. Piccolo for providing reagents. This work was supported by grants from the Associazione Italiana per la Ricerca sul Cancro (AIRC) and AIRC Special Program Molecular Clinical Oncology '5 per mille' (Grant no. 10016) and Italian Ministry of Health, to G.D.S., S.B. and A.R. M.M. is supported by the FIRB RBAP11Z4Z9 project from the Italian Ministry of Education and the FCT Investigator Programme IF/00694/2013 from the Portuguese Foundation for Science and Technology (FCT), Portugal. M.F. is supported by FIRB RBAP11T3WB and ERC grant no. 670126 Denovostem.

## Author contributions

G.S., N.R., A.Z., E.I., R.B., C.M. and C.N. performed the experiments. A.R. and E.C. performed mice experiments. M.M. performed the high-content screening. S.B. and M.F. performed bioinformatic analysis. G.S. and G.D.S. designed experiments and wrote the manuscript.
