## [Peer Review File · Nature Communications]

Reviewers' comments:

Reviewer #1 (expert in GR/breast cancer)(Remarks to the Author):

A. Summary of the key results

Activation of the glucocorticoid receptor with a variety of steroidal compounds was found via a chemical screen to result in increased YAP levels, nuclear accumulation and transcriptional activity of YAP. The activation is hypothesized to be indirect - through an induced cytoskeleton-dependent Yap activation, which is proposed to lead to chemoresistant cancer "stem" cells.

B. Originality and interest: if not novel, please give references

Dex has been shown to induce fibronectin previously, leading to increased hepatocyte spheroid formation (Please see: Abu-Absi SF1, Hu WS, Hansen LK Tissue Eng. Dexamethasone effects on rat hepatocyte spheroid formation and function. 2005 Mar-Apr;11(3-4):415-26.)
Glucocorticoids/GR activation have been shown to induce EMT genes in TNBC.

Thus the originality/novelty of this paper lies in the requirement for YAP expression, downstream of dex treatment, on stem cell formation, however the mechanism of YAP activation by fibronectin, and how YAP mediates colony formation is not determined in sufficient detail, nor is evidence of stem cell expansion actually demonstrated (spheroid formation is not equivalent)

C. Data & methodology: validity of approach, quality of data, quality of presentation.

The induction of YAP mRNA and protein appears to be a downstream effect of GR activation based on the siRNA experiments and demonstration of this indirect effect is adequate. However, the mechanism connecting cytoskeletal reorganization to YAP nuclear localization and phosphorylation is not clear to this reviewer, nor are the YAP-regulated target genes implicated in the ultimate spheroid identified.

In addition, the quality of the data is adequate but is redundant, for example, Figure

The data is quite redundant in parts, for example, Fig 5B could replace several of the Western blots shown early in the paper that demonstrate YAP protein induction with steroids. The FAK and pSrc-Y416 induction could be shown earlier and the YAP nuclear induction with siGR and siLATS shown in a single panel with these data.

D. Appropriate use of statistics and treatment of uncertainties

Statistics are adequate

E. Conclusions: robustness, validity, reliability

The conclusions of spheroid formation is increased by glucocorticoids in association with nuclear YAP translocation appear valid, but the conclusion that the spheroids indicate increased stem cell activity caused by YAP activity is overstated. Rather YAP activity appears to be required for this phenomenon.

F. Suggested improvements: experiments, data for possible revision

The experiments could be presented in a much more streamlined fashion, with fewer panels and the data consolidated into the most important experiments.

G. References: appropriate credit to previous work?

See above, a reference concerning spheroid formation and glucocorticoids in hepatocytes has been published previously.

H. Clarity and context: lucidity of abstract/summary, appropriateness of abstract, introduction and conclusions

The clarity is adequate, though overstated conclusions about stem cell expansion is made on the basis of spheroids from cell lines. The equation of spheroids with stem cell expansion is problematic given the lack of evidence that stem cells were formed, rather spheroids were counted.

Reviewer #2 (Expert in YAP/Breast cancer)(Remarks to the Author):

In this manuscript the authors present evidence to suggest that glucocorticoids induce expression of fibronectin that in turn activates focal adhesion kinase/Src signaling to decrease YAP phosphorylation and promote nuclear translocation. The authors identified the connection between glucocorticoids and YAP activation using a *in vitro* screen aimed at identifying upstream regulators of the Hippo signaling pathway. The study is performed well and the conclusions are well supported.

My concerns relate to the experimental evidence that support the overall model proposed by the authors.

The authors show that there is time lag between GR and YAP activation. It would be important for the authors to analyze changes in LATS kinase activity and YAP phosphorylation at multiple time points after BM stimulation. This would provide critical insight into the order of events and provide additional support for the author's model.

The authors should determine if BM induced YAP occurs when cells are held in suspension? If the model is correct the cells in suspension should not be able to initiate the FN-mediated signaling and hence not activate YAP, however, should retain GR activation. However, if the authors were to observe YAP activation in suspension, it would identify alternative or complementary pathways.

The clinical significance analysis should be performed using datasets from multiple studies to increase confidence.

Reviewer #3 (Expert in Actin remodeling/Src-FAK)(Remarks to the Author):

Summary: The authors performed a microscope-based screen with 640 clinically-used compounds and find that compounds acting via Glucocorticoid (GC) Receptor induced the activation of the YAP transcription factor (via decreased canonical LATS phosphorylation and nuclear localization) in MDA-MB-231 breast carcinoma cells. The GC signaling linkage to YAP is indirect and occurs over a relatively slow time frame (12-24 h). Betamethasone (BM) was the most potent and enhanced MDA-MB-231 colony size in soft agar in a GC-, YAP- and RU486 dependent manner.

Data-mining of published CIP-seq datasets was used to identify an enrichment of genes associated with cell adhesion in GC-stimulated cells. One of these, fibronectin (FN) mRNA and protein was elevated by BM stimulation of cells. Relatively high levels of an anti-integrin RGD peptide, changes in focal adhesion formation, FAK-Src phosphorylation were used as support for a FN signaling connection to YAP activation. Evidence was provided that GC stimulation could enhance secondary mammosphere formation of breast carcinoma cells as a measure of cancer stem cell expansion and an overlap between GC- and YAP-activation gene signatures was identified in basal-like breast cancer.

Opinion: The association of GC stimulation and a connection to YAP activation is interesting. However, the support for a specific mechanistic connection to YAP via FN signaling is weak - it is very difficult to prove the importance of one indirect gene induction linkage over another. FN could be one of several non-cell autonomous mechanisms affecting YAP activity 12-24 h after GC stimulation. The use of RGD peptide inhibition, changes in focal adhesion or actin stress fiber formation is not state of the art for interrogating the role of integrin signaling. Additionally, it is unclear how the changes in FN-adhesion signaling are connected to GC-enhanced 3D mammosphere growth where cells are not making canonical focal adhesion contacts in soft agar.

Overall, although the study is well written and experiments include appropriate controls, the overall impact and potential connections to cancer, stem cell renewal, or chemoresistance remain unclear.

Specific points:

1. Further mechanistic insights are needed to connect the signaling via focal adhesions in 2D with the role of integrins within growing mammospheres.
2. The term "chemo-resistant" and "cancer stem cells" are being loosely applied to results with alternate interpretations. Both GC and YAP can activate common (FN) and distinct gene targets. Is one more important than another? If so, provide additional proof of causality - FN, FAK, or ILK knockdown. What about the use of a FAK inhibitor?
3. Statement of page 7 "results prove that GR signaling sustains YAP oncogenic activity in breast cancer cells." This statement is not directly supported by the data. The data (Fig. 2) show that YAP expression is required for GR-enhanced colony growth. It is not shown that YAP oncogenic activity (effects of a YAP S5A mutant) requires GR input.
4. The colony growth values in Figure 2 are those above 100 μm . Effects of GC are on colony size and not necessarily colony initiation. Is this consistent with a stem cell hypothesis?
5. Fig. 4F-J. The data using 0.5 mg/ml RGD peptide addition for 6 h is not necessarily specific for blocking FN. It is the amount of detergent insoluble amount of FN that has the greatest signaling activity. At lower concentrations, RGD peptides have the strongest inhibitory effect on $\beta 3$ integrin signaling. The blots in panel H and images in panel J need improvement.
6. Fig. 5. The increase in vinculin-positive adhesions needs to be quantified. Are these sites of FN and integrin-specific signaling and tyrosine phosphorylation events? FAK pY925 and Src pY416 phosphorylation are rapid and transient events upon FN adhesion. There are questions as to whether long-term treatment with BM leads to these specific events as the data presented could be better. Using latrunculin A at 0.5 μM is like using a sledge hammer on cells. The increase in f-actin staining is of unclear significance and should be quantified. Can the authors differentiate tension generation from adhesion signaling?
7. Fig. 6. This reviewer would want to know specific methods used for secondary colony formation assay. How were cells prepped and harvested? Was viability checked? Is enumeration of colonies above 100 μm ? Is FN playing a signaling role in this assay?

RESPONSE TO REVIEWERS

Reviewer #1 (expert in GR/breast cancer)(Remarks to the Author):

A. Summary of the key results

Activation of the glucocorticoid receptor with a variety of steroidal compounds was found via a chemical screen to result in increased YAP levels, nuclear accumulation and transcriptional activity of YAP. The activation is hypothesized to be indirect - through an induced cytoskeleton-dependent Yap activation, which is proposed to lead to chemoresistant cancer "stem" cells.

B. Originality and interest: if not novel, please give references

Dex has been shown to induce fibronectin previously, leading to increased hepatocyte spheroid formation (Please see: Abu-Absi SF1, Hu WS, Hansen LK Tissue Eng. Dexamethasone effects on rat hepatocyte spheroid formation and function. 2005 Mar-Apr;11(3-4):415-26.) Glucocorticoids/GR activation have been shown to induce EMT genes in TNBC.

Thus the originality/novelty of this paper lies in the requirement for YAP expression, downstream of dex treatment, on stem cell formation, however the mechanism of YAP activation by fibronectin, and how YAP mediates colony formation is not determined in sufficient detail, nor is evidence of stem cell expansion actually demonstrated (spheroid formation is not equivalent)

C. Data & methodology: validity of approach, quality of data, quality of presentation.

The induction of YAP mRNA and protein appears to be a downstream effect of GR activation based on the siRNA experiments and demonstration of this indirect effect is adequate. However, the mechanism connecting cytoskeletal reorganization to YAP nuclear localization and phosphorylation is not clear to this reviewer, nor are the YAP-regulated target genes implicated in the ultimate spheroid identified.

We thank the reviewer for these useful comments.

The reviewer considers adequate the demonstration that YAP is an indirect downstream effector of GR, although he/she raised the following concerns: 1) the mechanisms of how YAP is activated by Fibronectin is not determined in sufficient detail; 2) the mechanisms connecting the cytoskeletal actin reorganization to YAP nuclear localization are not clear; 3) spheroids cannot be formally considered as "cancer stem cells"; 4) YAP-regulated target genes implicated in spheroid formation are not identified.

Concerning point 1 (mechanisms of how YAP is activated by Fibronectin is not determined in sufficient detail):

In our MS we propose that GR activation leads to strong induction and extracellular deposition of fibronectin, which in turn leads to increased formation of focal adhesions and concomitant activation of FAK/Src intracellular signalling. These activities end up with the induction of actin polymerization and increase of cell spreading, which are both well-established YAP upstream activators.

In particular, we showed that:

- a) Glucocorticoids induce expression of FN1 mRNA and release of FN1 protein.
- b) Increased FN1 deposition associates with increased focal adhesion number; increased FAK/Src activation; increased cell spreading.
- c) For epistasis, we show that inhibition of GR activation (RU486 treatment), inhibition of Fibronectin-

Integrin interaction (RGD treatment); inhibition of Src activation (Dasatinib/Saracatinib treatment) and inhibition of Actin polymerization (Latrunculin treatment), all prevent YAP activation upon glucocorticoids treatment.

To further detail the molecular axis governing YAP activation downstream of GR stimulation and to reinforce the link between focal adhesions and YAP following GR activation, we performed new experiments. In particular

- As also suggested by the reviewer 2, we show that cells grown in suspension and unable to make canonical focal adhesions are refractory to YAP activation upon GC treatments, while they are still able to activate FN1 expression (New Supplementary Figure 4f).
- As suggested by the reviewer 3 we used a specific FAK inhibitor in cells grown in 2D culture and we found that this treatment prevents actin polymerization and YAP activation downstream of GR (New Figure 5c and 5d and New Supplementary Figure S4c and S5).
- Moreover, as suggested by the reviewer 3, we performed new experiments with FN1 and Integrin α V (ITGAV) siRNA and we show that FN1 or ITGAV knockdown prevented actin polymerization and YAP activation downstream of GR (New Supplementary Figure 3h).

Concerning point 2 (the mechanisms connecting the cytoskeletal actin reorganization to YAP nuclear localization are not clear):

We agree with the comment since our MS does not provide insights into this issue.

We identified an unpredicted link between glucocorticoid receptor and YAP and we unravelled the molecular circuit connecting these two proteins and its biological effects. Therefore, our focus was not to investigate how the actin cytoskeleton controls YAP, rather to understand how the GR pathway intersects YAP signalling. In this context, we found that GR activation controls a general and very well-known YAP regulator, i.e. the actin cytoskeleton.

Indeed, although the literature linking the actin cytoskeleton to YAP activation is abundant (just few examples: Dupont et al Exp cell res 2016, Gaspar and Tapon, curr op cell biol 2014, Low FEBS 2014, Halder Nat rev mol cell biol 2012), the molecular mechanisms behind are not completely understood leaving this biological question as a major black box of this field.

Thus, while we share the reviewer's interest on these mechanisms, we think that their investigation goes beyond the scope of this MS.

Concerning point 3 (spheroids cannot be formally considered as "cancer stem cells")

The reviewer raises a concern about the use of mammosphere assay to monitor stem cell expansion ("spheroid formation is not equivalent" and "overstated conclusion as about stem cell expansion is made on the basis of spheroids from cell lines").

We agree with the reviewer comments. In the previous version of the MS, we employed standard and broadly used protocols of mammosphere assay to quantify the self-renewal activity of stem/early progenitor's mammary cells *in vitro* (Dontu et al 2003; Ponti et al. 2005; Shaw et al 2012; Kim et al. 2015). Moreover, we monitored the amount of CD44+/CD24- cells, a population of cells with high self-renewal potential and chemo-resistant (Al-Hajj et al 2003; Sheridan et al 2006; Phillips et al 2006), upon glucocorticoids treatment. Indeed, although in the breast cancer field the efficiency of mammosphere formation and the frequency of CD44+/CD24- cells are broadly considered a reliable way to monitor the self-renewal capacity of cancer cells (a key feature of cancer stem cells, Al-Hajj et al 2003; Ponti et al. 2005), these *in vitro* assays cannot be used to formally define and quantify the so-called Tumor-initiating cells/Cancer Stem Cells (O'Brian et al. 2010).

To formally prove the ability of glucocorticoids to sustain the self-renewal of cancer stem cells, we performed a golden standard bioassay monitoring the frequency of cancer stem cells: the limiting-dilution assay (O'Brian et al. 2010). We implanted serial dilution (from 125,000 to 15,000) of MDA-MB-231-shCTL or MDA-MB-231-shGR cells into the mammary fat pad of mice. 21 days after cell injection, tumour engraftment was analysed by palpation. As shown in new Figure 6e and 6f the engraftment of tumours from MDA-MB-231-shGR was strongly compromised when the number of injected cells was 15,000. On the contrary, MDA-MB-231-shCTL cell engraftments were similarly successful at all dilutions. As expected, the levels of YAP and its target SLUG were reduced in tumours from MDA-MB-231-shGR cells. Thus, based on these experiments we conclude that GR signalling is instrumental to maintain breast cancer stem cells traits.

Concerning point 4 (YAP-regulated target genes implicated in spheroid formation are not identified)

Concerning the downstream genes implicated in spheroid formation we identified SLUG among the Glucocorticoid-induced genes in MDA-MB-231 cells. Interestingly, SLUG is one of the most critical genes involved in self-renewal of normal and cancer breast cells and has been found to sustain spheroid growth (Medici et al 2008). SLUG is a known YAP target gene (Shao et al 2014), therefore we hypothesized that SLUG could mediate, at least in part, the effects of Glucocorticoids on spheroid formation downstream of YAP. Indeed, we confirmed that BM treatment induces SLUG upregulation (New Figure 6c). Moreover, we found that knocking down YAP in BM treated cells completely prevented SLUG induction, proving that YAP mediates the effect of glucocorticoids on SLUG (New Figure 6c). Finally, we demonstrated that SLUG induction is functionally required for BM-induced spheroid formation (New Figure 6d). This evidence provides insights on the mechanism of spheroid formation as requested by the reviewer.

In addition, the quality of the data is adequate but is redundant, for example, Figure 5B. The data is quite redundant in parts, for example, Fig 5B could replace several of the Western blots shown early in the paper that demonstrate YAP protein induction with steroids. The FAK and pSrc-Y416 induction could be shown earlier and the YAP nuclear induction with siGR and siLATs shown in a single panel with these data.

We agree with the reviewer, however, we think that showing the pSrc and pFAK induction earlier would affect the *consecutio logica* of the entire MS. Indeed, following the flow of the paper, we test the requirement of the FAK-Src pathway for the regulation of YAP by glucocorticoids only after having demonstrated the involvement of Fibronectin (Figure 4) downstream of steroids. Therefore, we think that showing pSrc/pFAK in early figures would be confusing for the readers.

D. Appropriate use of statistics and treatment of uncertainties

Statistics are adequate

E. Conclusions: robustness, validity, reliability

The conclusions of spheroid formation is increased by glucocorticoids in association with nuclear YAP translocation appear valid, but the conclusion that the spheroids indicate increased stem cell activity caused by YAP activity is overstated. Rather YAP activity appears to be required for this phenomenon. We agree with the reviewer; the sentence is not formally correct. Therefore, we now changed the title of the paragraph from "Glucocorticoids induce YAP-dependent cancer stem cells expansion and chemoresistance" to "YAP is required for Glucocorticoids-induced stem cell traits in breast cancer cells".

F. Suggested improvements: experiments, data for possible revision

The experiments could be presented in a much more streamlined fashion, with fewer panels and the data consolidated into the most important experiments.

Please see the answer above.

G. References: appropriate credit to previous work?

See above, a reference concerning spheroid formation and glucocorticoids in hepatocytes has been published previously.

The reference has been included in the text.

H. Clarity and context: lucidity of abstract/summary, appropriateness of abstract, introduction and conclusions

The clarity is adequate, though overstated conclusions about stem cell expansion is made on the basis of spheroids from cell lines. The equation of spheroids with stem cell expansion is problematic given the lack of evidence that stem cells were formed, rather spheroids were counted.

Please see the answer above.

Reviewer #2 (Expert in YAP/Breast cancer)(Remarks to the Author):

In this manuscript the authors present evidence to suggest that glucocorticoids induce expression of fibronectin that in turn activates focal adhesion kinase/Src signaling to decrease YAP phosphorylation and promote nuclear translocation. The authors identified the connection between glucocorticoids and YAP activation using a *in vitro* screen aimed at identifying upstream regulators of the Hippo signaling pathway. The study is performed well and the conclusions are well supported.

My concerns relate to the experimental evidence that support the overall model proposed by the authors.

The authors show that there is time lag between GR and YAP activation. It would be important for the authors to analyze changes in LATS kinase activity and YAP phosphorylation at multiple time points after BM stimulation. This would provide critical insight into the order of events and provide additional support for the author's model.

We thank the reviewer for the important suggestion. To address this point, we performed a time-course treatment of MDA-MB-231 cells with betamethasone and monitored LATS activation (by using phosphospecific antibody) and YAP phosphorylation on Ser127, which is a known phosphorylation site targeted by LATS. Interestingly, the new Supplementary Figure 2d shows that LATS inactivation is evident after 3h of BM treatment and preceded YAP de-phosphorylation. This result is in line with data shown in Figures 4 and suggests that YAP activation by GR is mediated by indirect mechanisms involving actin cytoskeleton and LATS.

The authors should determine if BM induced YAP occurs when cells are held in suspension? If the model is correct the cells in suspension should not be able to initiate the FN-mediated signaling and

hence not activate YAP, however, should retain GR activation. However, if the authors were to observe YAP activation in suspension, it would identify alternative or complimentary pathways. In agreement with the reviewer's comment, we think this is an important experiment to prove the involvement of cell adhesion for the GC-induced activation of YAP. To determine whether the BM-induced activation of YAP occurs when cells are not able to make adhesions to fibronectin, as suggested by the reviewer, we detached MDA-MB-231 cells by trypsin and maintained cells in suspension with or without glucocorticoid treatment for different times. As shown in new Supplementary figure 4F, while FN1 was induced by betamethasone, the levels of YAP and its target gene ANKRD1 did not increase in these experimental conditions, further supporting a crucial role of FN1 in YAP activation downstream of GR activation.

The clinical significance analysis should be performed using datasets from multiple studies to increase confidence.

We definitely agree with the reviewer and indeed, to increase the confidence level of our analyses, we assembled a compendium of gene expression data from thousands of primary breast tumours derived from different studies. While been represented by a single data matrix, the meta-dataset is not a single dataset, but rather a compendium of multiple studies. Specifically, as described in Material and Methods, the meta-dataset of breast cancer gene expression profiles used in this manuscript has been created starting from a collection of 4,640 samples from **27 major studies** (i.e., **independent data sets**) comprising microarray data of breast cancer samples annotated with histological tumor grade and clinical outcome (Enzo et al., 2015; Zanconato et al., 2015). The collection was normalized and annotated with clinical information as described in ref. Enzo et al., 2015. This resulted in a compendium comprising **3,661 unique** primary human breast cancers **from 25 independent cohorts**.

Reviewer #3 (Expert in Actin remodeling/Src-FAK)(Remarks to the Author):

Summary: The authors performed a microscope-based screen with 640 clinically-used compounds and find that compounds acting via Glucocorticoid (GC) Receptor induced the activation of the YAP transcription factor (via decreased canonical LATS phosphorylation and nuclear localization) in MDA-MB-231 breast carcinoma cells. The GC signaling linkage to YAP is indirect and occurs over a relatively slow time frame (12-24 h). Betamethasone (BM) was the most potent and enhanced MDA-MB-231 colony size in soft agar in a GC-, YAP- and RU486 dependent manner.

Data-mining of published CIP-seq datasets was used to identify an enrichment of genes associated with cell adhesion in GC-stimulated cells. One of these, fibronectin (FN) mRNA and protein was elevated by BM stimulation of cells. Relatively high levels of an anti-integrin RGD peptide, changes in focal adhesion formation, FAK-Src phosphorylation were used as support for a FN signaling connection to YAP activation. Evidence was provided that GC stimulation could enhance secondary mammosphere formation of breast carcinoma cells as a measure of cancer stem cell expansion and an overlap between GC- and YAP-activation gene signatures was identified in basal-like breast cancer.

Opinion: The association of GC stimulation and a connection to YAP activation is interesting. However, the support for a specific mechanistic connection to YAP via FN signaling is weak - it is very difficult to

prove the importance of one indirect gene induction linkage over another. FN could be one of several non-cell autonomous mechanisms affecting YAP activity 12-24 h after GC stimulation.

We thank the reviewer for his positive evaluation of our results. About the connection of GR to YAP via FN signalling, we agree with his comment. Indeed, the activation of GR controls several aspects of cell physiology and induces a plethora of genes involved in biological processes ranging from cellular metabolism to actin cytoskeleton and inflammatory response. Therefore, the effect of glucocorticoids on YAP could not be restricted only to FN1 induction and actin cytoskeleton remodelling. However, among several possible biological effects following GR stimulation, we have identified at least one crucial mechanism of YAP activation demonstrating that the adhesion-mediated actin cytoskeleton remodelling is required for the GR-dependent induction of YAP.

Indeed, we demonstrate that YAP activation downstream of GR is efficiently prevented by inhibiting :

- i) cell adhesion by several approaches, such as by RGD peptide treatment, by growing cells in suspension, by FN1 or ITGAV knockdown;
- ii) Src/FAK intracellular signalling (by Dasatinib/Saracatinib and FAK inhibitor treatment);
- iii) Actin polymerization (by latrunculin treatment).

In this MS we have investigate this interplay in breast cancer cells in particular. Considering the plethora of biological effects linked to these hormones, it is conceivable that in other tissues the GR - YAP connection could involve also other mechanisms. However, this goes beyond the scope of this MS and will be the subject of future investigations.

The use of RGD peptide inhibition, changes in focal adhesion or actin stress fiber formation is not state of the art for interrogating the role of integrin signaling.

We found several recent papers using RGD to inhibit fibronectin-induced signal (Mężyk-Kopeć et al, 2015 ; Ibuka et al 2015; Saito et al 2015; Cohen et al. 2014). However, as suggested by this reviewer, to strengthen the role of Fibronectin/Focal adhesions in the GC-induced activation of YAP we performed experiments using a specific and commonly used FAK inhibitor (PF-573228). As expected treatment with PF-573228 efficiently prevented actin polymerization, YAP nuclear accumulation and transcriptional activation upon GC treatment. These results have been introduced in Figure 5c and 5d and New Supplementary Figure S4c and S5.

Moreover, we have also performed experiments with cells in suspension (as suggested by reviewer #2). MDA-MB-231 cells have been detached by trypsin, maintained in suspension, and treated with glucocorticoids for different times. As shown in new Supplementary Figure 4f, in this experimental set-up betamethasone treatment led to induction of FN1. However, as expected, YAP protein levels and its functional activation, as judged by the expression of his direct target gene ANKRD1, were prevented. This evidence further supports the crucial role of cell adhesion in YAP activation downstream of GR activation.

Finally we have also performed experiments of FN1 and Integrin α V (ITGAV) siRNA transfection. As shown in new Supplementary Figure 3h, also FN1 or ITGAV knockdown prevented actin polymerization and YAP activation downstream of GR.

Additionally, it is unclear how the changes in FN-adhesion signaling are connected to GC-enhanced 3D mammosphere growth where cells are not making canonical focal adhesion contacts in soft agar.

The Reviewer is asking how cells growing in mammospheres, which cannot make canonical focal adhesions, could be able to activate YAP downstream of GR. Actually, we used the mammosphere assay to quantify YAP-dependent self-renewal changes induced by treatments provided during 2D culture conditions. For better understanding, the description of the assay has been improved both in the Results as well as in the Methods sections.

Briefly, to test whether the treatment with glucocorticoids (GCs) and other chemicals was associated with changes in the self-renewal activity, we adopted the procedure by Weinberg/Lander described in Gupta et al Cell 2009. Cells were maintained in 2D culture conditions, under which they were subjected to a 5-days treatment with glucocorticoids and/or other chemicals. Under these growing conditions cells **are completely proficient in activating the adhesion signalling**. To quantify self-renewal activity changes occurred during the treatment, after the 2D growing period, cells were subjected to mammosphere growing conditions.

Overall, although the study is well written and experiments include appropriate controls, the overall impact and potential connections to cancer, stem cell renewal, or chemoresistance remain unclear.

About potential connection to cancer please see answer below, point 4.

Specific points:

1. Further mechanistic insights are needed to connect the signaling via focal adhesions in 2D with the role of integrins within growing mammospheres.

Please see answer above.

2. The term "chemo-resistant" and "cancer stem cells" are being loosely applied to results with alternate interpretations. Both GC and YAP can activate common (FN) and distinct gene targets. Is one more important than another? If so, provide additional proof of causality - FN, FAK, or ILK knockdown. What about the use of a FAK inhibitor?

As suggested by the reviewer, we used a FAK inhibitor and found that this drug prevented YAP activation by glucocorticoids in cells grown in 2D, thus confirming the role of the Fibronectin/FAK/Src axis in YAP induction (New Figure 5c and 5d and New Supplementary Figure S4c and S5).

Moreover, we tested the effect of Glucocorticoids in cells treated by trypsin and maintained in suspension, which obviously cannot activate the FN signalling. As shown in New Supplementary Figure S4f, in these experimental setting glucocorticoids completely failed to activate YAP, while retained the ability to activate FN1 expression.

Finally, we performed new experiments with FN1 and Integrin αV (ITGAV) siRNA and we show that FN1 or ITGAV knockdown prevented actin polymerization and YAP activation downstream of GR (New Supplementary Figure 3h).

3. Statement of page 7 "results prove that GR signalling sustains YAP oncogenic activity in breast cancer cells." This statement is not directly supported by the data. The data (Fig. 2) show that YAP expression is required for GR-enhanced colony growth. It is not shown that YAP oncogenic activity (effects of a YAP S5A mutant) requires GR input.

The sentence has been removed from the MS.

4. The colony growth values in Figure 2 are those above 100 um. Effects of GC are on colony size and not necessarily colony initiation. Is this consistent with a stem cell hypothesis?

Actually, we used the soft agar colony formation assay only with the aim to test the oncogenic potential of GR activation. In order to properly address the question about its impact on tumor growth we have performed *in vivo* experiments by injecting MDA-MB-231 cells in which the expression of GR has been knock down by shRNA into the mammary gland fat pad of nude mice. In line with the *in vitro* results, we observed a consistent reduction of tumour growth and YAP levels and activity from tumours generated from GR knock down cells (now presented in Figure 6e-f). Based on the power of the *in vivo* experiments we decided to remove the soft agar assay from results. To support the evidence on the role of glucocorticoids in sustaining the self-renewal capacity of cancer stem cells, we performed the limiting-dilution assay, a golden standard bioassay to monitor the frequency of cancer stem cells (O'Brian et al. 2010). We injected serial dilution (from 125,000 to 15,000) of MDA-MB-231-shCTL or MDA-MB-231-shGR cells into the fat pad of mice. 21 days after cell injection, tumour engraftment was analysed by palpation. As shown in new Figure 6e and 6f and as previously mentioned, the engraftment of tumours from MDA-MB-231-shGR was strongly compromised when the number of injected cells was 15,000. On the contrary, MDA-MB-231-shCTL cell engraftments were similarly successful at all dilutions. As expected, the levels of YAP and its target SLUG were reduced in tumours from MDA-MB-231-shGR cells.

5. Fig. 4F-J. The data using 0.5 mg/ml RGD peptide addition for 6 h is not necessarily specific for blocking FN. It is the amount of detergent insoluble amount of FN that has the greatest signaling activity. At lower concentrations, RGD peptides have the strongest inhibitory effect on beta3 integrin signaling. The blots in panel H and images in panel J need improvement.

Images in panel h and j have been improved.

6. Fig. 5. The increase in vinculin-positive adhesions needs to be quantified.

We now quantified the number of vinculin-positive foci and the results are shown in new Supplementary Figure 4a

Are these sites of FN and integrin-specific signaling and tyrosine phosphorylation events?

To answer this question, we performed immunostaining of pFAK and vinculin in GC-treated cells. As shown in new Supplementary Figure 4b the number of vinculin and pFAK positive foci was increased after BM treatment and the two specific signals co-localized.

FAK pY925 and Src pY416 phosphorylation are rapid and transient events upon FN adhesion. There are questions as to whether long-term treatment with BM leads to these specific events as the data presented could be better.

Based on our data, the increase of focal adhesion foci induced by glucocorticoids is dependent on the transcriptional activation of FN1 by GR, a process that requires 2/3 hours and is maintained at least until 24h (Figure 4a-c and Supplementary Figure 3f).

In line with this, after 6h of GC treatment, we find that FAK is activated at focal adhesions (Supplementary Figure S4b) and YAP is de-phosphorylated and localized into the nucleus (Supplementary Figure 2d and Figure 4a).

We also observed FAK activation and YAP nuclear accumulation at 24h of treatment (Figure 4a and

5b). This evidence is consistent with the fact that, in our experiments, cells are continuously exposed to GCs that possibly lead to multiple cycles of FN1 release and adhesion signalling activation.

Using latrunculin A at 0.5 μ M is like using a sledge hammer on cells.

We agree with the reviewer. We repeated the experiment in Figure 5f using a 10-times lower concentration of latrunculin. As shown in new Supplementary Figure 5, at this concentration latrunculin was still able to prevent YAP activation by betamethasone.

The increase in f-actin staining is of unclear significance and should be quantified.

We now quantified the number of cells with stress fibres and the results are shown in new Supplementary Figure S4d.

Can the authors differentiate tension generation from adhesion signaling?

In our experimental set up, we always observed an impact on tension generation upon manipulation of the adhesion signalling. As reported in new Supplementary Figure 5, we blocked the adhesion signalling at different levels, by FN1 inhibition (RGD treatment), FAK inhibition (PF treatment), and Src inhibition (Dasatinib treatment), thus always preventing the increase in stress fibres formation induced by BM. Hence, we conclude that the actin cytoskeleton remodelling induced by GC is mediated by the adhesion signalling.

7. Fig. 6. This reviewer would want to know specific methods used for secondary colony formation assay. How were cells prepped and harvested? Was viability checked? Is enumeration of colonies above 100 μ M? Is FN playing a signaling role in this assay?

For the mammospheres assay, established protocols were adopted (Dontu et al., 2003; Rustighi et al., 2014). In brief, cells were treated with the indicated drugs for 5 days in 2D culture. Then, cells were trypsinized, stained with trypan blue to check cell viability and plated as single cell suspension on ultra-low attachment plates (Corning) in mammosphere growing medium (Dontu et al., 2003; Rustighi et al., 2014). When spheres reached $>100 \mu$ m size, they were counted, harvested with p1000, washed with PBS, pelleted and dissociated with trypsin. Single cells were stained with trypan blue, counted and reseeded for a second round of mammosphere formation.

About the role of FN in this assay please see answer above.

Reviewers' comments:

Reviewer #1 (Remarks to the Author):

These authors have concluded that they have identified a link between glucocorticoid receptor and YAP activation that requires direct GR activation of Fibronectin (FN1) gene expression.

1. The discovery of fibronectin activation as a target of GR activation is not new, being first reported in Cell in 1983, and in several publications since then, limiting the overall novelty of the paper.

More references:

Regulation of fibronectin biosynthesis by glucocorticoids in human fibrosarcoma cells and normal fibroblasts.

Oliver N et al. Cell. 1983 May;33(1):287-96.

Dexamethasone-Mediated Activation of Fibronectin Matrix Assembly Reduces Dispersal of Primary Human Glioblastoma Cells.

Shannon S et al.

PLoS One. 2015 Aug 18;10(8):e0135951. doi: 10.1371/journal.pone.0135951. eCollection 2015.

The subsequent connection to YAP activation as the authors' state in their rebuttal letter is established by various previous papers and the authors use various inhibitors and shRNAs in this known pathway to confirm these established connections. This reviewer was expecting more original mechanistic insights might be made about the mechanisms connecting actin reorganization and YAP activity yet via GR dependency, yet the authors state this is beyond the scope of the MS.

2. The authors have stated in the rebuttal that the major originality of their paper is in the role of YAP in breast cancer stem cell activity. The first submission made this statement on the basis of in vitro spheroid assays. While the in vivo experiments in the resubmission show that in a single cell line there is less TNBC cell line tumor uptake in mice with GR shRNA-expressing cells versus control shRNA expressing cells, GR depletion is most certainly affecting a myriad of target genes besides FN1 (and Yap activity) to result in altered tumor uptake in this dilution model. GR knockdown alone is not a suitable experiment to assess the role of YAP in stem cell activity in vivo. To implicate YAP activity in this GR-mediated phenotype, activated versus inactive YAP would have to be repleted to see if it rescues the phenotype.

In summary, the experiments as designed do not justify the conclusions about GR-mediated YAP activation in cancer stem cells, and the connection of GR activity to fibronectin activity, and therefore in turn to YAP, are not mechanistically surprising.

Reviewer #2 (Remarks to the Author):

The reviewers have addressed my concerns raised during the first review. However, I would like the authors to include the data in Supp Fig 2d and Supp fig 4f as part of the main figures as these are critical data that support the overall conclusion of the manuscript.

Reviewer #4 (Remarks to the Author):

In this work the authors describe how a screen of FDA approved drugs identified glucocorticoids, such as betamethasone (BM), as promoting YAP expression. The authors subsequently document that this is the result of increased fibronectin (FN) expression leading to elevated signaling through Src and alteration of YAP phosphorylation. Finally, data is presented that this pathway promotes 'stemness' in breast cancer cell lines. I was not one of the original reviewers, but have been asked to comment on the revised manuscript. As such, I will endeavor to restrict my comments to issues raised by the original reviewer.

Major issues previously raised

1 Lack of detailed mechanism linking glucocorticoid receptor (GR) to YAP – The authors do propose the framework of a mechanism that is based on GR binding the FN promoter and subsequent activation of integrin signaling. Their proposal is clear, but should be backed up with additional data. To be thorough, the authors should demonstrate GR ChIP on the FN promoter in the cells that they are using. They should test whether FN over-expression is sufficient to activate YAP and whether FN siRNA prevents BM from activating YAP. The authors should also probe whether Src-family kinases and FAK are activated following BM treatment and perform western blot analysis of pMST1/2 and pLATS1/2 following BM treatment +/- the various inhibitors. I agree with the authors that a complete solution regarding how actin tension regulates YAP is beyond the scope, but testing phosphorylation of key hippo pathway components is not difficult and the pLATS blot in Figure S2 is too low quality to be informative.

2 Quality of the stemness assays – The authors have made some solid attempts to improve these. Nonetheless, to be rigorous they should test whether the stemness surrogates that they use in vitro are dependent on FN, and if FN over-expression can promote stemness in their cell line models.

3 Discussion – The authors should discuss whether there is any data from the millions of patients who have received glucocorticoids that support their proposal of increased YAP activity and mammary stem cell function (at the moment there is only vague discussion on cancer incidence). The data in figure 5 only demonstrates a correlation between GR expression and YAP target genes, it does not demonstrate that externally varying glucocorticoid signaling affects YAP target genes. The authors should also mention at the start the common use of RU486 as an anti-progesterone drug. The original Dupont et al paper on YAP and mechano-transduction also showed some effects of TAZ, can the authors explain why TAZ is not affected by GR signaling.

4 Suitability of topic for Nature Communications – I think if the authors were to rigorously demonstrate their model then the work would be suitable.

5 Technical – I could not easily locate appropriate controls for off-target effects when using siRNA, such as the use of multiple independent sequences or rescue with siRNA-resistant cDNA. These must be provided.

Reviewer#1 comments

These authors have concluded that they have identified a link between glucocorticoid receptor and YAP activation that requires direct GR activation of Fibronectin (FN1) gene expression.

1. The discovery of fibronectin activation as a target of GR activation is not new, being first reported in Cell in 1983, and in several publications since then, limiting the overall novelty of the paper.

More references:

Regulation of fibronectin biosynthesis by glucocorticoids in human fibrosarcoma cells and normal fibroblasts.

Oliver N et al. Cell. 1983 May;33(1):287-96.

Dexamethasone-Mediated Activation of Fibronectin Matrix Assembly Reduces Dispersal of Primary Human Glioblastoma Cells.

Shannon S et al.

PLoS One. 2015 Aug 18;10(8):e0135951. doi: 10.1371/journal.pone.0135951. eCollection 2015.

The subsequent connection to YAP activation as the authors' state in their rebuttal letter is established by various previous papers and the authors use various inhibitors and shRNAs in this known pathway to confirm these established connections.

The GR-FN1 axis, which has been already reported as also mentioned by the reviewer, does not represent our key finding. In this manuscript it turned out to be a link in a more complicated and unprecedented chain we have focused on and that represents the genuine discovery of our work: namely, the mechanism controlling GR-induced cancer stem cell expansion via YAP. Thus, we argue about how the GR-FN1 axis can limit the novelty of our paper and mostly about how a few dots in a background can be considered established connection without having experimentally proved it. Connecting dots is very easy, only after having obtained results.

We want again to stress that the mechanism we have highlighted involves: GC-dependent GR transcriptional activation; GR-dependent FN1 induction and deposition; FN1-dependent Src/FAK activation; Src/FAK-dependent F-actin polymerization and cytoskeleton remodelling; LATS-dependent YAP stabilization/nuclear accumulation/transcriptional activation; YAP-mediated Slug induction; YAP- and Slug-dependent Cancer Stem Cells expansion; GR- and YAP-dependent induction of chemoresistance.

Thus, in our MS we provide a number of previously unknown insights on the biological activity of GR in breast cancer. In particular, we have shown that:

- i) GR activates YAP;
- ii) GR activates Src;

- iii) GR inhibits the main Hippo pathway kinase LATS;
- iv) GR induces Slug expression as indirect effect of YAP activation;
- v) GR sustains mammosphere growth in vitro and is required for tumour engraftment in vivo;
- vi) YAP mediates the activity of GR on chemoresistance;
- vii) Inhibition of GR with FDA-approved RU486 blocks the oncogenic activities of YAP in breast cancer stem cells.

Moreover, it is worth noting that, as references for the GR-FN1 connection, the reviewer has cited works focused on fibroblasts and patient-derived glioblastoma. Our results have been obtained from a completely different biological context, namely Triple Negative Breast Cancer cells. Considering that Glucocorticoids exert a plethora of different biological activities, in different tissues, we think that a direct FN1 activation by GR was not obvious at all in TNBC cells.

Along with this consideration, our decision to focus on FN1 and actin cytoskeleton was not driven by the evidence in literature, rather from an unbiased analysis of GR target genes in the context of a TNBC cells dataset. Thus, our experimental evidence point on FN1 as mediator of GR on YAP.

This reviewer was expecting more original mechanistic insights might be made about the mechanisms connecting actin reorganization and YAP activity yet via GR dependency, yet the authors state this is beyond the scope of the MS.

Again, our decision to focus on FN1 and actin cytoskeleton was not driven by the evidence in literature, rather from unbiased analysis of GR target genes in the context of TNBC.

Moreover, as already commented in our first rebuttal, the focus of our MS is not to investigate how the actin cytoskeleton controls YAP, which is a big and still open biological question of this field, rather to understand how the GR pathway intersects YAP signalling. As for many other laboratories worldwide, our research has been focused, since years, on studying the connection between the actin cytoskeleton and YAP, therefore we are aware that the identification of original mechanistic insights on this topic would require an entire MS.

2. The authors have stated in the rebuttal that the major originality of their paper is in the role of YAP in breast cancer stem cell activity. The first submission made this statement on the basis of in vitro spheroid assays. While the in vivo experiments in the resubmission show that in a single cell line there is less TNBC cell line tumor uptake in mice with GR shRNA-expressing cells versus control shRNA expressing cells, GR depletion is most certainly affecting a myriad of target genes besides FN1 (and Yap activity) to result in altered tumor uptake in this dilution model. GR knockdown alone is not a suitable experiment to assess the role of YAP in stem cell activity in vivo. To implicate YAP activity in this GR-mediated phenotype, activated versus inactive YAP would have to be repleted to see if it rescues the phenotype.

In summary, the experiments as designed do not justify the conclusions about GR - mediated YAP activation in cancer stem cells, and the connection of GR activity to fibronectin activity, and therefore in turn to YAP, are not mechanistically surprising.

About the alleged statement of ours that the major originality of our paper is the role of YAP in breast cancer stem cell activity we respectfully disagree: we never stated that. Indeed, we consider the ability of glucocorticoids to activate YAP and the effects of Glucocorticoid Receptor signalling on cancer stem cells as the major originality of our paper.

About the in vivo experiment, we performed the golden standard assay to monitor cancer stem cells number, as already commented in the first rebuttal.

In particular, we designed the limiting dilution assay to address the specific and correct concern of this reviewer that mammospheres are not formally equivalent to stem cells. Thus, to demonstrate that GR signalling is crucial for cancer stem cells maintenance and tumour initiation, we knocked-down GR in MDA-MB-231 cells injected in mice and monitored tumour engraftment.

In our in vitro experiments, we have extensively and carefully demonstrated that the GR-induced effects are dependent on YAP (Fig.1d, Fig.1e; Fig.3g; Fig.6b; Fig.6c; Fig.6h; Fig.7b; Fig.7c). Indeed, no this reviewer nor the others raised concerns about this issue during the first evaluation. Thus, we are really puzzled that this reviewer is questioning the dependence of our results on YAP and we think that our experiment was well planned in order to answer the specific concern of this reviewer about the “stem cells” traits induced by glucocorticoids and not about the dependence of GR activity on YAP.

Reviewer#2 comments

The reviewers have addressed my concerns raised during the first review. However, I would like the authors to include the data in Supp Fig 2d and Supp fig 4f as part of the main figures as these are critical data that support the overall conclusion of the manuscript.

We thank the reviewer for the very useful suggestions which led us to strongly improve the quality of our MS and for his/her positive evaluation of our additional experiment. We agree with him and we moved the supplementary Figures 2d and 4f in the main figures.

Reviewer #4 (Remarks to the Author):

In this work the authors describe how a screen of FDA approved drugs identified glucocorticoids, such as betamethasone (BM), as promoting YAP expression. The

authors subsequently document that this is the result of increased fibronectin (FN) expression leading to elevated signaling through Src and alteration of YAP phosphorylation. Finally, data is presented that this pathway promotes 'stemness' in breast cancer cell lines. I was not one of the original reviewers, but have been asked to comment on the revised manuscript. As such, I will endeavor to restrict my comments to issues raised by the original reviewer.

Major issues previously raised

1 Lack of detailed mechanism linking glucocorticoid receptor (GR) to YAP – The authors do propose the framework of a mechanism that is based on GR binding the FN promoter and subsequent activation of integrin signaling. Their proposal is clear, but should be backed up with additional data. To be thorough, the authors should demonstrate GR ChIP on the FN promoter in the cells that they are using.

We thank the reviewer for considering clear the mechanism that we propose.

As the reviewer requested, we performed ChIP experiments to confirm that upon Betamethasone (BM) treatment in MDA-MB-231 cells (the triple negative breast cancer cell line used in our experiments) GR binds FN1 promoter. The result that is presented in new Supplementary figure S3d, clearly shows that BM treatment induces about 10-fold increase of GR binding to FN1 promoter.

They should test whether FN over-expression is sufficient to activate YAP and whether FN siRNA prevents BM from activating YAP.

About the overexpression, there are very few papers about FN1 ectopic expression since FN1 is a large protein (over 200kDa) and its expression is difficult. We think that siRNA of FN1 and other inhibitors that we have used should be enough to prove the question.

About FN1 siRNA, instead, we agree with the reviewer. In fact, in our original MS we do show that siFN1 and siTGAV prevented YAP activation upon BM treatment (Supplementary Figure S3i). To strengthen this result and to address the last concern of this reviewer (controls for off-targets), we performed the same experiment using additional independent siRNAs sequences (Supplementary Figure S3h and S3j).

The authors should also probe whether Src-family kinases and FAK are activated following BM treatment and perform western blot analysis of pMST1/2 and pLATS1/2 following BM treatment +/- the various inhibitors. I agree with the authors that a complete solution regarding how actin tension regulates YAP is beyond the scope, but testing phosphorylation of key hippo pathway components is not difficult and the pLATS blot in Figure S2 is too low quality to be informative.

We agree with the reviewer that measuring Src and FAK activation upon BM treatment is a crucial control to demonstrate that glucocorticoids activate the focal adhesion/Src pathway. Indeed, in our original MS we do show that Src and FAK are activated upon BM treatment (by western blot of phosphorylated Src and FAK, Figure 5b), and RU486 co-treatment prevented their activation.

To improve the quality of pLATS blot (previously showed in Supplementary Figure 2d), we run again the same lysates used for the original figure and performed a new western blot. Now the pLATS blot is significantly improved in quality and clearly shows that BM inhibits LATS-1 activation (New figure 3g).

Moreover, as requested by the reviewer, we performed western blot analysis of pLATS1 and pMST1/2 upon BM treatment following treatments with dasatinib and FAK inhibitor, in order to investigate the possible involvement of the Hippo pathway in BM-induced YAP activation. As shown in new Supplementary Figure S4f, MST1 and pMST1/2 are not affected by BM treatment, suggesting that GR activates YAP in a LATS-dependent, but MST-independent manner. These results are in line with other works showing that, although the role of MST1/2 in the Hippo pathway has been clearly established, MST1/2 are not essential for LATS1/2 activation under various conditions such as GPCR stimulation or E-cadherin signalling (Yu et al Cell 2012; Yu et al. Gene and Dev 2013; Zhao et al. Gene and Dev 2012; Kim et al PNAS 2011).

2 Quality of the stemness assays – The authors have made some solid attempts to improve these. Nonetheless, to be rigorous they should test whether the stemness surrogates that they use in vitro are dependent on FN, and if FN over-expression can promote stemness in their cell line models.

Regarding FN1 overexpression please see our comment above.

We agree with the reviewer and, as he/she requested, we performed new mammosphere experiment upon siFN1 transfection and BM treatment. As shown in new Supplementary figure S6d, knockdown of FN1 efficiently prevented the capacity of BM to sustain mammosphere formation.

3 Discussion – The authors should discuss whether there is any data from the millions of patients who have received glucocorticoids that support their proposal of increased YAP activity and mammary stem cell function (at the moment there is only vague discussion on cancer incidence). The data in figure 5 only demonstrates a correlation between GR expression and YAP target genes, it does not demonstrate that externally varying glucocorticoid signaling affects YAP target genes.

About this point, we agree with the reviewer that the analysis of YAP target genes in the tumour tissues of patients receiving glucocorticoids, could provide a valuable confirmation of our results. Unfortunately, despite the large amount of data available in public repositories from patients receiving glucocorticoids, we could not find any useful data obtained from the mammary tissue, and in particular from tumours of TNBC patients. Since in our work we demonstrated an association between GR and YAP specifically in the mammary tissue, in particular in TNBC cells and considering that GC may have very different biological effects in different tissues, we think that using data from other tissues to further demonstrate the GR/YAP axis would be inconsistent with our experimental evidences and would not be necessary at this stage.

The authors should also mention at the start the common use of RU486 as an anti-progesterone drug.

We have now mentioned it in the main text.

The original Dupont et al paper on YAP and mechano-transduction also showed some effects of TAZ, can the authors explain why TAZ is not affected by GR signaling.

Our findings, similar to other studies (Cordenonsi, Cell 2011; Zhu. Dev Cell 2016; Lehmann, Nat Comm 2016), unveil a significant functional difference of the major Hippo-pathway effectors YAP and TAZ. This may be explained by several reasons, and among them the major structural differences between YAP and TAZ which could be required for putative GC-induced post-translational modifications or for specific protein-protein interactions. Thus, our results confirmed the existence of divergent role of the major Hippo-pathway effectors YAP and TAZ.

In a recent publication, performing mass spec experiments to identify all the YAP and TAZ interacting proteins, Wang et al. (Mol Cell Proteomics 2014) identified hundreds of YAP and TAZ interacting proteins, which belong to several different signalling pathways. Strikingly, only 34% of TAZ-interacting proteins identified by Wang were also able to interact with YAP. This evidence strongly suggests the existence of several potential differences of YAP and TAZ regulation, which still remain to be determined. Our evidence that glucocorticoids activate YAP, but not TAZ, suggests that this hormonal signal could activate parallel, and still unknown, pathways, which are able to specifically prevent TAZ (but not YAP) activation under condition that normally would activate it (mechanical tensions). Further investigations of our laboratory will be focused on explaining this interesting issue. In this regard, we added the following comment in the discussion section of the paper: "Our findings, as also reported in other studies [16,64,65], demonstrate the existence of a functional difference of the Hippo-pathway effectors YAP and TAZ. The major structural differences between YAP and TAZ and their different repertoire of post-translational modification and interacting proteins [66] might

explain the divergent response of these two proteins to GR stimulation.”

4 Suitability of topic for Nature Communications – I think if the authors were to rigorously demonstrate their model then the work would be suitable.

5 Technical – I could not easily locate appropriate controls for off-target effects when using siRNA, such as the use of multiple independent sequences or rescue with siRNA-resistant cDNA. These must be provided.

For many targets, as for FN1/Integrin and FAK in addition to siRNA we also used specific inhibitors (RGD, PF-573228), for YAP we used siRNA, inhibitor (verteporfin) and overexpression of a dominant nuclear form of YAP.

For the most relevant gene in our study, Glucocorticoid receptor, we used 1 siRNA and 1 shRNA (different targeted sequences) and well established inhibitor (RU486). However, to address the reviewer’s concern, we added experiments with additional independent siRNA sequences for FN1; ITGAV; and YAP (Supplementary Figures S3h-i and S6b).

REVIEWERS' COMMENTS:

Reviewer #4 (Remarks to the Author):

The authors have made reasonable attempts to address the issues raised. It is slightly disappointing that they have not managed to find more pertinent clinical data, but on balance I would favor publication.

Point by point response.

REVIEWERS' COMMENTS:

Reviewer #4 (Remarks to the Author):

The authors have made reasonable attempts to address the issues raised. It is slightly disappointing that they have not managed to find more pertinent clinical data, but on balance I would favor publication.

We thank the reviewer for the helpful suggestions, for considering adequate our responses to his/her concerns, and for finally considering our ms suitable for publication.